# Long-Lead-Time Prediction of Storm Surge Using Artificial Neural Networks and Effective Typhoon Parameters: Revisit and Deeper Insight

**Wei-Ting Chao** [1], **Chih-Chieh Young** [1,2,*], **Tai-Wen Hsu** [1,3,*], **Wen-Cheng Liu** [4] and **Chian-Yi Liu** [5]

[1] Center of Excellence for Ocean Engineering, National Taiwan Ocean University, Keelung 20224, Taiwan; away19850624@gmail.com

[2] Department of Marine Environmental Informatics, National Taiwan Ocean University, Keelung 20224, Taiwan

[3] Department of Harbor & River Engineering, National Taiwan Ocean University, Keelung 20224, Taiwan

[4] Department of Civil and Disaster Prevention Engineering, National United University, Miaoli 36063, Taiwan; wcliu@nuu.edu.tw

[5] Center for Space and Remote Sensing Research, National Central University, Taoyuan 32001, Taiwan; cyliu@csrsr.ncu.edu.tw

* Correspondence: youngjay@ntou.edu.tw (C.-C.Y.); twhsu@mail.ntou.edu.tw (T.-W.H.)

**Abstract:** Storm surge induced by severe typhoons has caused many catastrophic tragedies to coastal communities over past decades. Accurate and efficient prediction/assessment of storm surge is still an important task in order to achieve coastal disaster mitigation especially under the influence of climate change. This study revisits storm surge predictions using artificial neural networks (ANN) and effective typhoon parameters. Recent progress of storm surge modeling and some remaining unresolved issues are reviewed. In this paper, we chose the northeastern region of Taiwan as the study area, where the largest storm surge record (over 1.8 m) has been observed. To develop the ANN-based storm surge model for various lead-times (from 1 to 12 h), typhoon parameters are carefully examined and selected by analogy with the physical modeling approach. A knowledge extraction method (KEM) with backward tracking and forward exploration procedures is also proposed to analyze the roles of hidden neurons and typhoon parameters in storm surge prediction, as well as to reveal the abundant, useful information covered in the fully-trained artificial brain. Finally, the capability of ANN model for long-lead-time predictions and influences in controlling parameters are investigated. Overall, excellent agreement with observations (i.e., the coefficient of efficiency CE > 0.95 for training and CE > 0.90 for validation) is achieved in one-hour-ahead prediction. When the typhoon affects coastal waters, contributions of wind speed, central pressure deficit, and relative angle are clarified via influential hidden neurons. A general pattern of maximum storm surge under various scenarios is also obtained. Moreover, satisfactory accuracy is successfully extended to a much longer lead time (i.e., CE > 0.85 for training and CE > 0.75 for validation in 12-h-ahead prediction). Possible reasons for further accuracy improvement compared to earlier works are addressed.

**Keywords:** storm surge; effective typhoon parameters; artificial neural networks; knowledge extraction method; long-lead-time prediction

## 1. Introduction

Storm surge, an abnormal rise of water driven by meteorological conditions, has been an important research topic since the 1950s [1] due to its devastating impact on coastal communities. In general,

coastal areas with rich natural resources and great economic potential become highly developed regions. However, the coastal environment might face threats from severe weather systems, e.g., storms, typhoons, or hurricanes [2]. Historical records have shown many catastrophic tragedies, e.g., the storm surge of 1953 [3] in the North Sea coastal area (2167 deaths) and the storm surge of 1970 in Bangladesh (300,000 deaths). In the last decade, Hurricane Katrina swept through the Gulf of Mexico, U.S. in 2005 and destroyed New Orleans city with 1836 deaths and $125 billion in economic losses [4]. Later in 2013, Typhoon Haiyan hit the Philippines and left 6000 people dead. In the future, more destructive hazards from storm surge would be expected, especially under the influence of global warming/climate change (i.e., record-breaking super typhoons) and growing population (up to 2–5 billion by 2080) in coastal areas over the world [5–7]. In order to achieve coastal hazard mitigation (i.e., early warning for a coming event as well as appropriate design of protective structures), continuous efforts have been made to better predict storm surge in real time and statistically assess its extreme variation [8–13].

Accurate and efficient prediction of storm surge plays an essential role in disaster management (either for early warning or protection planning). The basic idea for surge prediction is to capture the fundamental components [14]. Storm surge induced by typhoons mainly consists of: (i) wind setup (i.e., strong onshore wind causes significant rise of sea level) and (ii) pressure setup (e.g., a large pressure drop of 100 mb during a severe typhoon event leads to a 100-cm rise in sea level). In addition to atmospheric forcing, physical processes in the motions of the ocean (e.g., interactions with waves) and variation of coastal topography/geometry (e.g., influences of shoaling and resonance) would further complicate the magnitude of storm surge (e.g., a considerable rise in surge level) at a shallow-water study area. In order to estimate storm surge from these effects, various modeling tools have been developed and can be divided into three major groups: (i) empirical formulas [2], (ii) hydrodynamic models [15], and (iii) artificial intelligence approaches [16].

Tools of the first kind seek empirical procedures to combine different effects into a single parameter (or more) for the estimation of storm surge (e.g., [2,17–21]). Through statistical analysis (e.g., linear regression) of observation data, these earlier works obtained good agreement for the relationships between maximum surge height and typhoon characteristics (e.g., pressure deficit and wind speed) at given locations. Typically, this simple tool can account for half of total variability of storm surge (with correlation coefficient $CC \cong 0.6$). A log-relation between the surge level and its probability of occurrence has also been found and used for the prediction of extreme design surge [22].

The second group predicts storm surge numerically by using fluid dynamics and atmospheric driving forces [15]. Two-dimensional depth-integrated hydrodynamic models in the Cartesian structure grid system are commonly utilized to simulate storm surge, where governing equations are nonlinear shallow water equations with Boussinesq and hydrostatic approximations (e.g., [23]). The driving forces (i.e., pressure and wind fields) can be specified using parametric cyclone models [24,25] for hindcast/diagnostic purposes. ("Surge modeling is an art" [26]). To effectively obtain more realistic results, a great amount of effort has been devoted to the development (or improvement) of integrated surge modeling systems over the past decades [27]. For more detailed and updated progress, the reader is referred to the articles by Sheng et al. [9], Wu et al. [12], Torres et al. [13], Dietrich et al. [28], and Kohno et al. [29]. A brief review is given in the following.

In terms of atmospheric forcing, the real-time observation data, revised parametric cyclone models, and weather prediction models were implemented for accurate storm surge simulation [13,30–35]. As for hydrodynamic aspect, a variety of methods for treatment of irregular coastal boundaries were applied, including boundary-fitted transformed grid systems, nested and multiple grids, and unstructured grids ([36–42] among many others). While bottom shear stress might be overestimated (or underestimated) in two-dimensional models, fully three-dimensional models capable of resolving detailed vertical flow structure were developed to better predict storm surges over complicated coastal topography [43,44]. The baroclinic effect (i.e., effect of density stratification) on storm surges can also be solved [45]. Further, in addition to surge-tide interactions [37,40,46,47], the wave-induced effect was taken into account through coupling wind wave models [12,28,35,44,48–51].

To date, full-physics coupled models based on high-performance computational resources have shown significant improvement and successful applications for risk assessment of storm surge [8,28]. However, note that storm surge prediction is still sensitive to atmospheric forcing. The relative errors between simulated and measured water levels are less than 10% (or up to 50%) while the track and magnitude of typhoons are precisely (or poorly) described (e.g., see [13,35]).

The artificial intelligent (AI) approaches provide an alternative way for predicting storm surge [11,16,52–54] as well as tidal variation [55]. Among various AI branches, the most popular method over the past two decades were artificial neural networks (ANNs) which mimic the human brain to effectively learn complicated rules regarding natural phenomena from sufficient data [56]. Based upon learning, error tolerance, and generalization capability of ANNs (especially for highly nonlinear systems), extensive and successful applications in various fields (e.g., meteorology, hydrology, water resources, hydraulics, and coastal engineering) can be found in the literature ([10,21,57–61] among many others). Regarding storm surge predictions, explorations of ANN applications generally can be divided into three typical categories.

The earlier attempt was to achieve efficient and accurate short-term (one-hour-ahead) prediction through constructing an appropriate (optimal) ANN model that captures the nonlinear relationship among observed storm surge and meteorological conditions [16,21,62]. Based upon fast computations, incorporation of ANNs into an operational storm surge forecasting system has also been reported [63]. In general, the data-driven ANNs outperform statistical approaches and process-based models, e.g., *RMSE* (root-mean-square error) < 10 cm and CC (correlation coefficient) > 0.90 [64]. Another type of application is to predict storm surge using a dynamic and neural network hybrid model [57,65]. Similar to the concept of predictor and corrector, the two components complement each other with respect to their strengths and limitations in such a combination framework. Better agreement between (corrected) prediction and observation can be obtained. The reader is referred to some recent studies for similar ideas and deeper discussion [66,67]. The other method is to reproduce storm surges of synthetic and/or historical events based fully on the databases from previous simulations when measurement data at some relevant areas might be unavailable [11].

The experiences in past studies with a main focus on network structure (i.e., number of hidden neurons), data division, and training parameters have laid a strong foundation for accurate and efficient ANN-based storm surge modeling [16]. To develop ANN-based forecast models, a systematic approach has also been proposed [61]. However, two critical issues remain to be solved, i.e., (i) prediction lead-time and (ii) black-box feature. In the former problem, given predictions with an insufficient lead-time (e.g., one-hour in most previous works), effective coastal disaster management (prevention/mitigation) through preparedness and early warning systems would hardly be achieved. In the past decade, three-hour ahead storm surge predictions with a recursive approach were attempted [62], yet prediction skills in terms of lead time and accuracy were still quite limited. Recognizing the demand for practical decision-making process, Kim et al. [10] recently re-examined ANN approaches and showed successful predictions of after-runner storm surges on the Tottori coast, Japan in a much longer lead-time (e.g., 5, 12, 24 h). In their work, the ANN models were appropriately trained and tested to reproduce three representative (i.e., 100-year return period) surge events induced by severe typhoons (e.g., typhoon Maemi with a minimum central pressure of 910 hPa) moving in a similar track. For the areas with frequent typhoon invasions from variable paths (e.g., 10 major paths in Taiwan), however, accurate long-lead-time surge prediction (defined by a lead time up to 12 h in this study) is still challenging. For the latter, like the black box, complex mathematical processes inside the artificial neural networks are difficult to interpret [68,69] despite their capability for solving various nonlinear problems. To the best of our understanding, no satisfactory explanation for the behavior of ANNs has been offered in earlier coastal applications. Knowledge within these networks has not yet been clearly revealed.

In this paper, therefore, we revisit the topic of storm surge prediction using artificial neural networks and effective typhoon parameters with the main purpose of extending further applicability (i.e., the prediction lead time for early warning) and gaining a deeper insight (i.e., knowledge inside the

neural network). By analogy with the physical modeling approach, typhoon parameters are carefully examined and selected to develop the ANN-based storm surge model (for a range of lead-time from 1 to 12 h). A knowledge extraction method based on backward tracking and forward exploration procedures is also proposed to analyze the roles of hidden neurons and controlling parameters in storm surge prediction as well as to reveal abundant, useful information from the fully-trained artificial brain. Finally, the capability of the ANN model for long-lead-time predictions and the influences of controlling parameters are investigated and discussed.

## 2. Description of Study Site and Data Collection

Taiwan is an island country in Eastern Asia with a population of 23.7 million over a total area of 36,000 km$^2$. Surrounded by the East China Sea, Philippine Sea, South China Sea, and Taiwan Strait (120–122° E and 21–25.5° N), total coastline of this island is about 1566 km and the coastal areas are densely populated with major economic activity. Situated in the northwest Pacific Ocean, Taiwan suffers from four typhoons on average every summer and autumn season. Induced by the typhoon, the elevated sea water level (due to low central pressure and strong winds) together with heavy rainfall, abundant river runoff, and high tide often cause composite disaster (e.g., surges, coastal inundation, floods, and mudflows) that results in significant losses of life and property, e.g., losses of 4.8 billion New Taiwan dollars in agriculture and fisheries during Typhoon Haitang in 2005 and losses of 1.1 billion NT dollars after Typhoon Krosa in 2007).

In this paper, we chose the northeastern region of Taiwan as the study area, where the largest storm surge, over 1.8 m, during Typhoon Krosa (2007) has been observed in historical records. The tidal data of a representative station, i.e., Longdong, indicated by a red triangle in Figure 1a, over a period from 2005 to 2014 were collected from the Central Weather Bureau (CWB). Harmonic analysis [70] was then performed to obtain the residual non-tidal signal (i.e., storm surge). The surge deviation during Typhoon Krosa is shown in Figure 1b, where the thick gray line is the measured total water level; the thin gray line is the astronomical tide from harmonic analysis; and the thick black line represents storm surge. For the same period, a total of 13 historical typhoon events with tracks No. 2 and 3 (see Figure 1a) was also collected. Available information includes the track/path (i.e., center locations in latitude and longitude) and the intensity of each typhoon. Table 1 summarizes the central pressure ($P_c$), max. wind speed ($V_c$), and radius of typhoon ($R_7$) for the selected events, i.e., Haitang (2005), Talim (2005), Longwang (2005), Bilis (2006), Kaemi (2006), Sepat (2007), Krosa (2007), Kalmegi (2008), Fungwong (2008), Sinlaku (2008), Morakot (2009), Soala (2012), and Matmo (2014).

**Table 1.** Central pressure ($P_c$), maximum wind speed ($V_c$), and radius ($R_7$) of typhoons as well as Max. surge ($S$) in historical events.

| Name | Year | Track | $P_c$ (hPa) | $V_c$[1] (m/s) | $R_7$ (km) | Max. $S$ (cm) |
|---|---|---|---|---|---|---|
| Haitang [2] | | 3 | 912 | 55 | 280 | 103 |
| Talim | 2005 | 3 | 920 | 53 | 250 | 144 |
| Longwang | | 3 | 925 | 51 | 200 | 44 |
| Bilis | 2006 | 2 | 978 | 25 | 300 | 41 |
| Kaemi | | 3 | 960 | 38 | 200 | 15 |
| Sepat | 2007 | 3 | 920 | 53 | 250 | 20 |
| Krosa | | 2 | 925 | 51 | 300 | 183 |
| Kalmaegi | | 2 | 970 | 33 | 120 | 23 |
| Fungwong | 2008 | 3 | 948 | 43 | 220 | 43 |
| Sinlaku | | 2 | 925 | 53 | 280 | 53 |
| Morakot [2] | 2009 | 3 | 955 | 40 | 250 | 71 |
| Soala | 2012 | 2 | 960 | 38 | 220 | 39 |
| Matmo | 2014 | 3 | 960 | 38 | 200 | 11 |

[1] $V_c$ is 10-min sustained wind speeds. [2] Validation events.

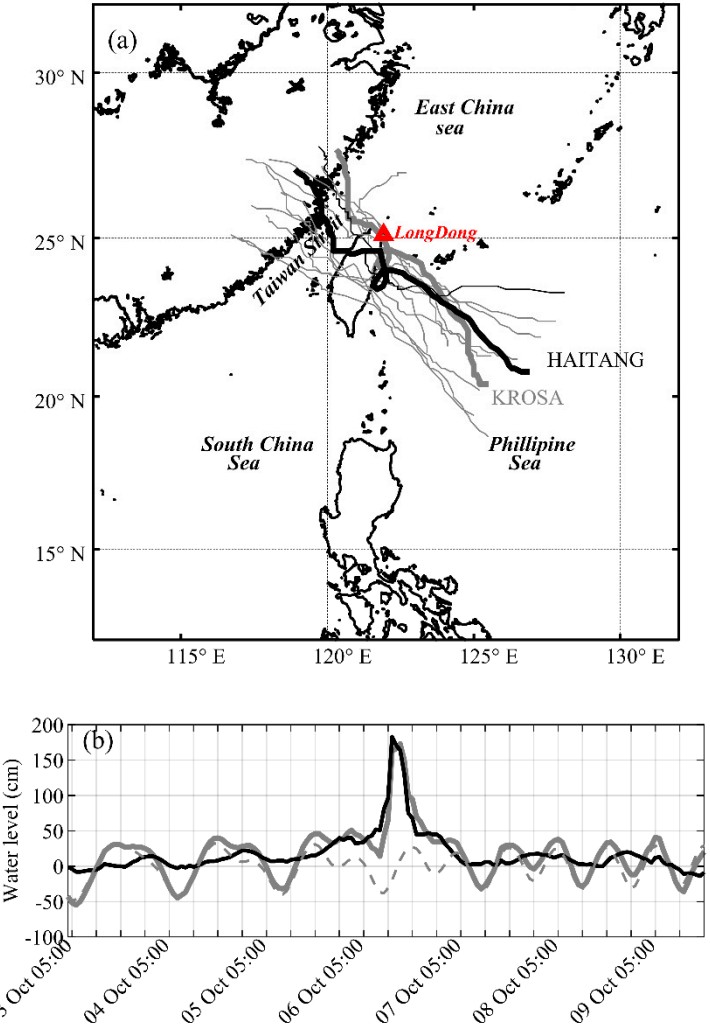

**Figure 1.** (**a**) Tracks of selected historical typhoons (2005–2014) and location of Longdong station; (**b**) total water level, astronomical tide, and storm surge deviation at Longdong during Typhoon Krosa.

## 3. BPNN Forecasting Model and Knowledge Extraction Method

### 3.1. Effective Controlling Parameters

Effective controlling parameters for predicting storm surges are characteristics of a typhoon. In general, the magnitude of storm surge at a tidal station increases with the central pressure deficit and maximum wind of the typhoon. From a physical point of view, storm surges can be described using nonlinear shallow water equations [15], i.e.,

$$\frac{\partial \eta}{\partial t} + \frac{\partial [U(h+\eta)]}{\partial x} + \frac{\partial [V(h+\eta)]}{\partial y} = 0, \tag{1}$$

$$\frac{\partial U}{\partial t} + U\frac{\partial U}{\partial x} + V\frac{\partial U}{\partial y} - fV = -g\frac{\partial \eta}{\partial x} - \frac{1}{\rho}\frac{\partial P_a}{\partial x} + F_x - B_x, \tag{2}$$

$$\frac{\partial V}{\partial t} + U\frac{\partial V}{\partial x} + V\frac{\partial V}{\partial y} + fU = -g\frac{\partial \eta}{\partial y} - \frac{1}{\rho}\frac{\partial P_a}{\partial y} + F_y - B_y, \tag{3}$$

where $x$, $y$, and $t$ represent the horizontal space and time, $\eta$ is the water surface elevation, $U$ and $V$ are the horizontal velocities, $P_a$ is the atmospheric pressure, $f$ is the Coriolis parameter, and $F$ and $B$ indicate the surface and bottom shear stresses, respectively. In practice, by substituting typhoon

characteristics into a parametric cyclone model, the atmospheric pressure $P_a$ and wind shear stress $F$ are specified. When we numerically solve the governing equations, discretization with finite difference method (or other methods) give the coefficient for each physical variable. Notice that both spatial and temporal increments should be chosen carefully to obtain desired accuracy and stability [71]. From the algebraic viewpoint, water elevation $\eta$ at the next time step can be further expressed as a function of $\eta$, $U$, and $V$. Similarly, horizontal velocities $U$ and $V$ can be expressed as a function of $P$ and $F$. Therefore, $\eta$ at next time step can be rearranged and written as

$$\eta^{n+1} = f\left(\eta, P_a, F_x, F_y\right) = f\left(\eta, P_a, W_x, W_y\right), \tag{4}$$

where $W_x$ and $W_y$ are the wind speeds in $x$ and $y$ directions, respectively.

In terms of efficient storm surge prediction using a neural network, a suitable set of input-output pairs are required and can be properly designed by analogy with the physical modeling approach above. In this study, the target output is the storm surge itself, instead of the total water level commonly used in earlier studies [16,62]. The benefit is that the artificial neural network can truly reflect the variation of storm surge rather than dominant tides [64]. The total water level can be obtained later by a simple linear superposition since nonlinear effects between surges and tides are mild (e.g., about 10% increase in total water level along the west coast) or even insignificant especially for the northeastern coastal waters of Taiwan [72].

For the inputs, previous works attempted to include all possible factors, e.g., local meteorological conditions and typhoon characteristics [62]. However, this may not be the best way for long lead-time storm surge prediction because only a relatively short-term variation can be reflected by local meteorological conditions. Local meteorological conditions are also not severe enough (or give redundant information) for storm surge generation when a typhoon is far away (or nearby). In particular, these ineffective inputs that lead to higher dimensions may hinder the learning capability and prediction performance of an artificial neural network.

By contrast, this study focuses on characteristics of a typhoon, i.e., the effective controlling parameters. In addition to storm surge ($S$), central pressure deficit ($\Delta P_c$), and maximum wind speed ($V_c$), we consider the location of a typhoon by its distance ($L$) and relative angle ($\theta_c$) rather than the longitude and latitude. Particularly, the relative angle of a typhoon (measured counterclockwise with 0 degree from the east) can imply the favorable wind direction (e.g., $\theta_w \cong \theta_c + 115°$) near a tidal station which is an important factor for storm surge prediction. Note that only the relative angle is needed here since a linear shift between $\theta_w$ and $\theta_c$ causes no difference for the artificial neural network model. The radius of typhoon ($R_7$) that reflects the affected area with wind speed over 15 m/s is also taken into account. Furthermore, the forward speed ($U_F$) and direction ($\theta_F$) are adopted to provide the information on possible impacts of a typhoon. Although high uncertainty exists in typhoon tracks, these dynamic factors estimated by the center locations between two successive observations would improve long-lead-time (12 h) surge predictions especially for mildly varied typhoons. Here, we also point out that the so-called invasion angle, i.e., the difference between typhoon moving direction and the line from tidal station to typhoon center, was used in Tseng et al. [62]. When a typhoon from any direction passes exactly through the tidal station, however, the resulting influences cannot be correctly described by the definition used in previous work (i.e., due to a value of zero for that angle). Overall, this study employs a total of eight effective controlling parameters to predict storm surge (see the depicted diagram in Figure 2a). Also, a series of combinations among these effective controlling parameters (see Table 2) is considered to examine their roles and contributions. Note that the values of effective typhoon parameters are fully based on the real observation. Although they can also be obtained from numerical weather predictions, accuracy and uncertainty of atmospheric models would be another critical issue. The geographical parameter is not considered here. However, the effects of coastal topography/geometry on surge level at a specific site can be implicitly included in the artificial neural network model. For various tidal stations, a general (or universal) approach for ANN models will be further investigated in the future.

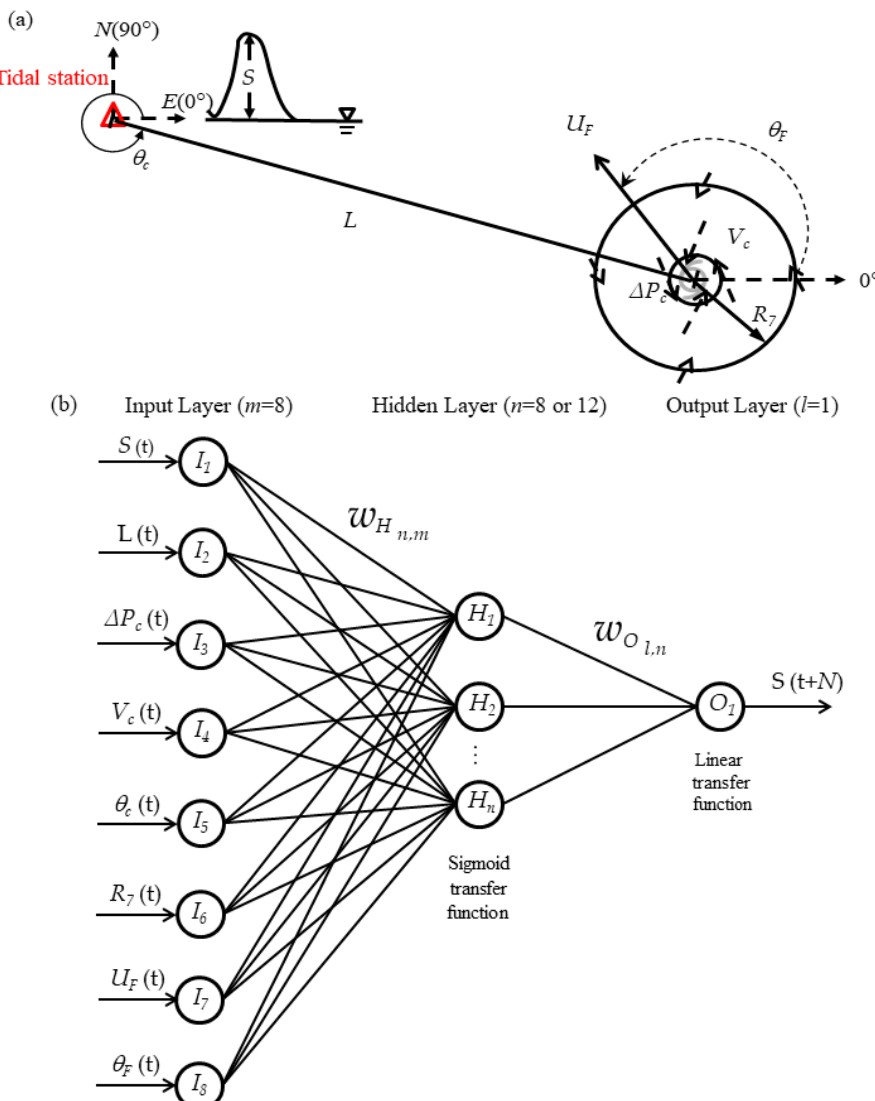

**Figure 2.** (**a**) Sketch of effective controlling parameters for storm surge prediction; (**b**) architecture of the back-propagation neural network.

**Table 2.** Six combinations of effective controlling parameters for storm surge prediction.

| Type | Input Parameters |
|---|---|
| S | $S(t)$ |
| SLP | $S(t), L(t), \Delta P_c(t)$ |
| SLV | $S(t), L(t), V_c(t), \theta_c(t)$ |
| SLPV | $S(t), L(t), \Delta P_c(t), V_c(t), \theta_c(t)$ |
| SLPVR | $S(t), L(t), \Delta P_c(t), V_c(t), \theta_c(t), R_7(t)$ |
| SPVLRF | $S(t), L(t), \Delta P_c(t), V_c(t), \theta_c(t), R_7(t), U_F(t), \theta_F(t)$ |

*3.2. Back-Propagation Neural Network (BPNN)*

The BPNN which consists of a number of neurons in three (i.e., input, hidden, and output) layers (see Figure 2b for the network structure) is utilized in this study. The inputs are normalized by their ranges, giving −1 and +1 for the minimum and maximum, respectively. Subsequently, each hidden or

output neuron receives a weighted sum of inputs from the previous layer and converts them into a temporary or final output signal through an activation function, i.e.,

$$H_n = f(w_{H_{n,m}}I_m + B_{H_m}) \text{ or } O_l = f(w_{O_{l,n}}H_n + B_{O_l}), \tag{5}$$

where $I_m$ is the normalized input of neuron $m$, $H_n$ is the temporary signal of the neuron $n$, $O_l$ is the final output of neuron $l$, $w_{H_{n,m}}$, $w_{O_{l,n}}$, $B_{H_m}$, and $B_{O_l}$ are the weights and biases, and the hyperbolic tangential sigmoid and linear transfer functions are adopted in the hidden and output layers, i.e., $f(x) = [2/(1 + e^{-2x})] - 1$ and $f(x) = x$, respectively.

Based on back-propagation of errors (i.e., $e_l = T_l - O_l$, where $T_l$ is the target), the training process of the neural network keeps updating the weights and biases to minimize the cost function $C_{NN}$ till required accuracy or maximum iterations, i.e.,

$$C_{NN} = \frac{1}{P}\sum_{p=1}^{P}\sum_{l=1}^{L} e_l^2(p) \tag{6}$$

where $P$ is the total number of inputs and $p$ indicates the index for the $p$-th input. To achieve fast (second-order) convergence, the Levenberg-Marquardt learning algorithm combining both Gauss–Newton and steepest descent approaches is used [73], i.e.,

$$\Delta w = -[H + \mu I]^{-1}G = [J^T J + \mu I]^{-1}J^T e \tag{7}$$

where $H = J^T J$ is the approximated Hessian matrix, $I$ is the identity matrix, $G = J^T e$ is the gradient, $J$ is the Jacobian matrix that contains first derivatives of the network error $e$ with respect to the weights and bias, and the parameter $\mu$ controls the magnitude and direction of each update.

Further, three different statistical indices are considered to evaluate the prediction performance, i.e., the root mean square error (*RMSE*), the coefficient of correlation (*CC*), and the coefficient of efficiency (*CE*)

$$RMSE = \sqrt{\frac{1}{N}\sum_{i=1}^{N}[(S_m)_i - (S_o)_i]^2}, \tag{8}$$

$$CC = \frac{\sum_{i=1}^{N}\left[(S_m)_i - \bar{S}_m\right]\left[(S_o)_i - \bar{S}_o\right]}{\sqrt{\sum_{i=1}^{N}\left[(S_m)_i - \bar{S}_m\right]\sum_{i=1}^{N}\left[(S_o)_i - \bar{S}_o\right]}}, \tag{9}$$

$$CE = 1 - \sum_{i=1}^{N}[(S_o)_i - (S_m)_i]^2 / \sum_{i=1}^{N}\left[(S_o)_i - \bar{S}_o\right]^2, \tag{10}$$

where $N$ is the total number of data points, $S_m$ and $S_o$ are the modeled and observed surges, respectively, and the overbar indicates the mean value.

### 3.3. Knowledge Extraction Method (KEM)

In this study, inspired by the generation and learning algorithm of the artificial neural network, we propose a knowledge extraction method based on backward tracking and forward exploration procedures to examine the contribution of effective controlling parameters to the prediction results.

In the backward tracking stage, the output neuron with a linear transfer function can be used to identify the most influential hidden neurons. Like the multiple regression analysis, the weight $w_O$ implies the responses of final output to the change in temporary signals $H_n$ (e.g., see Figure 5). The signs of $w_O$ and $H_n$ would determine whether the storm surge increases or decreases during the next n hours.

The bias $B_O$ can be viewed as an intercept value (i.e., a reference value). Similarly, the most influential hidden neurons can be further utilized to track the influences of each controlling parameter. However, one should notice the competition between weights $w_H$ and biases $B_H$. The variation of inputs would be hardly reflected in a strongly biased neuron due to the hyperbolic tangential sigmoid function, i.e., $H_n \cong +1/-1$. In other words, the nonlinear effect of one factor can be revealed unless the bias is mild or properly balanced by other weighted inputs.

For the forward exploration stage, we use a series of synthesis typhoon events to reveal the information and knowledge inside the brain of the artificial neural network (e.g., see Figure 6). Note that the input items are all the same but their values (combinations) should be appropriately specified. Initially, the storm surge at study area is assumed to be 0 m while the synthetic typhoon is at some distance away, e.g., 600 km. Subsequently, the synthetic typhoon moves with a typical forward speed, e.g., 15–20 km/h. For the path, a simple straight line commonly used for evaluating typhoon impacts is considered [74]. In other words, the tracks can be easily determined using different relative angles and center location at the landfall. Possible ranges of central pressure deficit (wind speed) and forward angle are determined according to historical records. It is also assumed that constant central pressure deficit (wind speed) is maintained during the passage of the synthetic typhoon. Based upon the scenarios above, the artificial neural network with repeated forward computations can readily indicate maximum storm surges under various conditions.

## 4. Results and Discussion

Storm surge prediction for Longdong, northeastern Taiwan, was conducted using artificial neural networks with eight effective typhoon parameters. During the period of 2005 to 2014, a total of 13 historical typhoons moving along tracks No. 2 and 3 were collected (see Table 1). Two typhoon events, i.e., Haitang (2005) and Morakot (2009) which induced large ($S > 100$ cm) and mild ($S > 50$ cm) storm surges respectively, were chosen for model validation while the others were used for training. The prediction lead time ranges from t + 1 to t + 12 h. To achieve reasonable accuracy without a common over-fitting issue, the hidden layer would employ eight neurons in all cases except for 12-h-ahead prediction with twelve neurons. The following Section 4.1 presents the excellent results of one-h-ahead predictions in both training and validation phases (see the scatter plots in Figures 8e and 11e). For the purpose of brevity, only two representative cases with significant storm surges ($S > 100$ cm) are shown here, i.e., Typhoon Krosa (training) and Haitang (validation). Note that the objective of a reliable one-h-ahead short-term prediction is not for early warning but to facilitate further analysis of the roles of hidden neurons and typhoon parameters using the backward tracking procedure of KEM in Sections 4.2 and 4.3. Five static typhoon parameters would be the focus and the dynamic/kinematic factors (i.e., forward speed ($U_F$) and angle ($\theta_F$)) will be discussed in long-lead-time prediction. Subsequently, by the exploration procedure of KEM noted in the foreword, Section 4.4 reveals useful information and knowledge inside the artificial neural networks. Finally, the capability of the ANN storm surge model for extended lead-time (i.e., 3, 6, 9, and 12 h) predictions as well as the influences of effective controlling parameters (see six combinations in Table 2) are investigated in Section 4.5. In particular, possible reasons for improvement in accuracy between the previous and present studies are discussed.

### 4.1. One-hour Ahead Prediction

Figure 3 presents the temporal variations of typhoon parameters (see (a) for forward speed and forward angle; (b) for distance and radius of storm; (c) for maximum wind speed and relative angle; (d) for central pressure deficit) and storm surge deviation during the training event, i.e., Typhoon Krosa. In the beginning, the location of the typhoon was at a distance of over 600 km away, southeast of Longdong station (i.e., relative angle about 300°). Krosa had a notably large radius of 300 km with a central pressure of 925 hPa (i.e., $\Delta P_c = 1013 - P_c = 88$ hPa) and maximum wind speed up to 51 m/s. Subsequently, Krosa moved northwest toward Taiwan with an average forward speed of 16.2 km/h.

During landfall, Krosa formed a looped track (see (a) for the sharp turns around 18:00 on 6 October 2007) due to the influence of island topography [75]. Meanwhile, the topography destroyed Typhoon Krosa, weakening its maximum wind speed and central pressure deficit. As for the coastal water, the mean sea level was apparently affected by the typhoon as its distance to Longdong station became smaller than its radius. A maximum surge up to 180 cm was generated when the nearest distance was 50 km away. Afterwards, surge deviation decreased gradually. Overall, the one-hour-ahead prediction and observation of surge (red line and black circle in Figure 3e) are in excellent agreement. The well-trained ANN model captures both peak surge and rising/falling pattern accurately with *RMSE* = 6.22 cm, CC = 0.988, and CE = 0.975 (see Table 3).

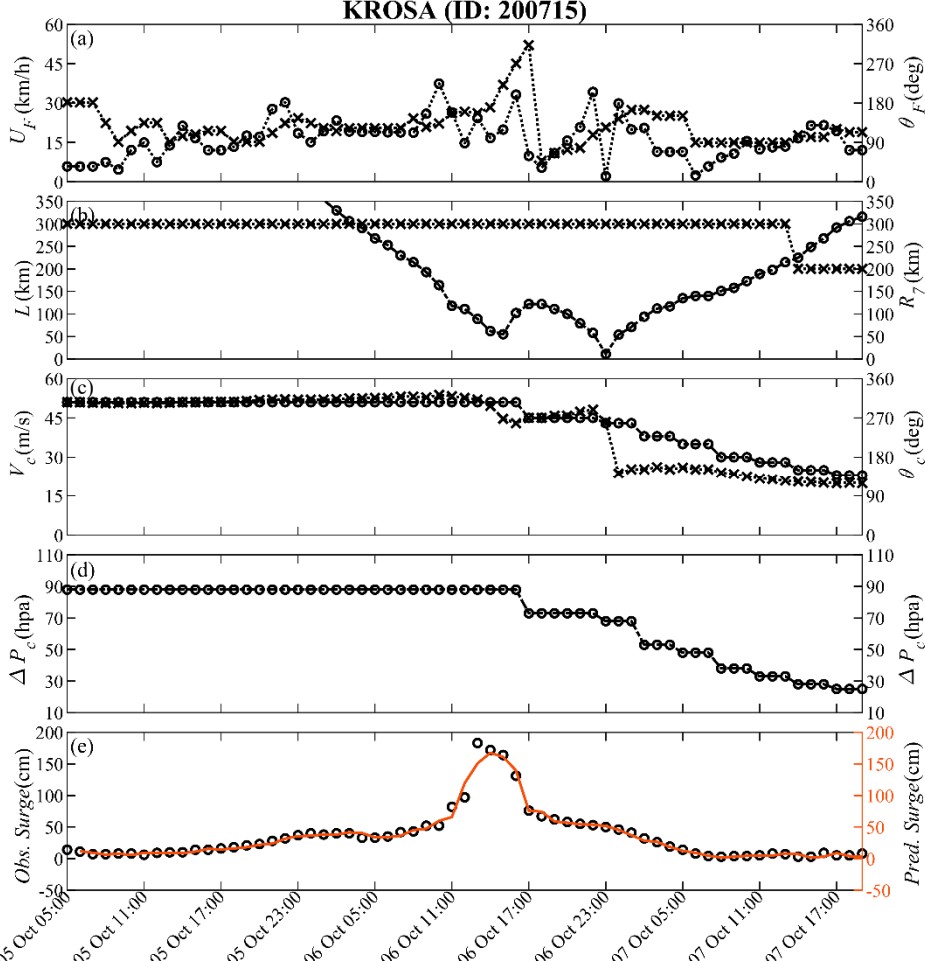

**Figure 3.** Temporal variations of typhoon parameters and storm surge at Longdong during Krosa event (training): (**a**) forward speed (circle) and forward angle (cross); (**b**) relative distance (circle) and radius of typhoon (cross); (**c**) wind speed (circle) and relative angle (cross); (**d**) central pressure deficit (circle); (**e**) observed (circle) and predicted (red lines) storm surges.

**Table 3.** Assessment of different lead-time surge predictions during Krosa event (training).

| | Lead Time (h) | | | | | | | | | | | | | | |
|---|---|---|---|---|---|---|---|---|---|---|---|---|---|---|---|
| | 1 | | | 3 | | | 6 | | | 9 | | | 12 | | |
| | *RMSE* | *CC* | *CE* | *RMSE* | *CC* | *CE* | *RMSE* | *CC* | *CE* | *RMSE* | *CC* | *CE* | *RMSE* | *CC* | *CE* |
| S | 11.06 | 0.961 | 0.923 | 24.62 | 0.804 | 0.625 | 35.76 | 0.586 | 0.229 | 38.55 | 0.541 | 0.125 | 41.81 | 0.536 | 0.03 |
| SLP | 11.14 | 0.960 | 0.922 | 20.87 | 0.876 | 0.731 | 27.17 | 0.850 | 0.572 | 26.96 | 0.813 | 0.555 | 28.55 | 0.792 | 0.535 |
| SLV | 7.97 | 0.980 | 0.961 | 12.97 | 0.954 | 0.896 | 13.83 | 0.965 | 0.884 | 18.61 | 0.911 | 0.796 | 19.61 | 0.907 | 0.780 |
| SLPV | 7.89 | 0.980 | 0.960 | 12.56 | 0.952 | 0.902 | 12.71 | 0.965 | 0.902 | 16.98 | 0.950 | 0.830 | 18.45 | 0.921 | 0.805 |
| SLPVR | 7.05 | 0.984 | 0.968 | 8.16 | 0.979 | 0.958 | 10.49 | 0.968 | 0.933 | 11.46 | 0.961 | 0.922 | 15.63 | 0.928 | 0.860 |
| SLPVRF | 6.22 | 0.988 | 0.975 | 7.12 | 0.984 | 0.968 | 10.01 | 0.969 | 0.939 | 10.27 | 0.969 | 0.937 | 14.95 | 0.938 | 0.872 |

Figure 4 shows similar results for the validation event, i.e., Typhoon Haitang. The central pressure, maximum wind speed, and radius of Haitang were 910 hPa, 55 m/s and 280 km, respectively. When Haitang approached Taiwan, the distance between its center location and Longdong station (about 125 km) was not as close as that of Krosa. Despite its severe intensity, the generated surge only reached a maximum of 103 cm. The developed ANN model also demonstrates good generalization capability for the validation event. The comparison between prediction and observation yields *RMSE* of 8.85 cm, CC of 0.955, and CE of 0.908 (see Table 4).

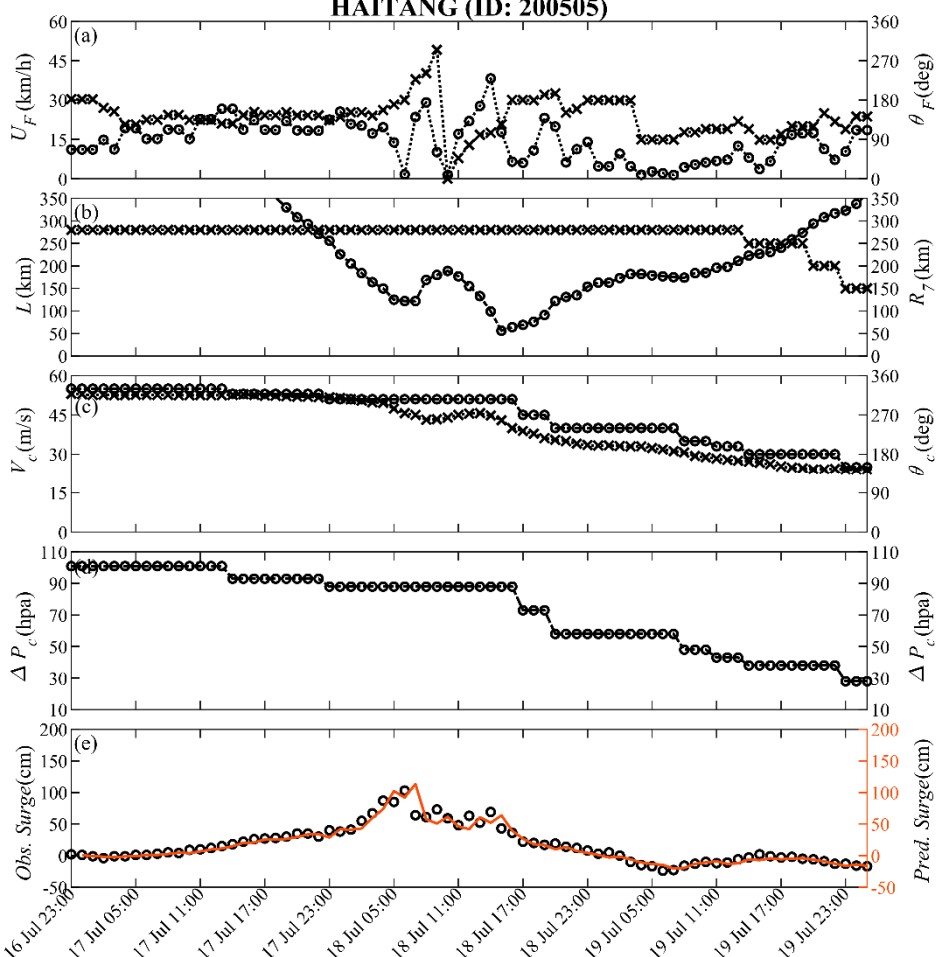

**Figure 4.** Temporal variations of typhoon parameters and storm surge at Longdong during Haitang event (validation): (**a**) forward speed (circle) and forward angle (cross); (**b**) relative distance (circle) and radius of typhoon (cross); (**c**) wind speed (circle) and relative angle (cross); (**d**) central pressure deficit (circle); (**e**) observed (circle) and predicted (red lines) storm surges.

**Table 4.** Assessment of different lead-time surge predictions during Haitang event (validation).

| | Lead Time (h) | | | | | | | | | | | | | | |
|---|---|---|---|---|---|---|---|---|---|---|---|---|---|---|---|
| | **1** | | | **3** | | | **6** | | | **9** | | | **12** | | |
| | *RMSE* | *CC* | *CE* | *RMSE* | *CC* | *CE* | *RMSE* | *CC* | *CE* | *RMSE* | *CC* | *CE* | *RMSE* | *CC* | *CE* |
| S | 9.86 | 0.948 | 0.886 | 16.92 | 0.853 | 0.670 | 19.75 | 0.751 | 0.561 | 23.76 | 0.647 | 0.385 | 28.50 | 0.472 | 0.147 |
| SLP | 9.79 | 0.954 | 0.887 | 16.44 | 0.884 | 0.689 | 17.02 | 0.854 | 0.674 | 17.94 | 0.816 | 0.654 | 18.14 | 0.809 | 0.645 |
| SLV | 8.58 | 0.958 | 0.913 | 14.60 | 0.897 | 0.754 | 16.75 | 0.865 | 0.684 | 17.37 | 0.852 | 0.671 | 17.47 | 0.850 | 0.670 |
| SLPV | 7.66 | 0.966 | 0.931 | 13.62 | 0.907 | 0.786 | 15.74 | 0.903 | 0.721 | 16.35 | 0.868 | 0.708 | 16.75 | 0.851 | 0.705 |
| SLPVR | 7.41 | 0.967 | 0.934 | 13.52 | 0.898 | 0.789 | 14.66 | 0.936 | 0.758 | 15.54 | 0.915 | 0.736 | 15.80 | 0.874 | 0.737 |
| SLPVRF | 8.85 | 0.955 | 0.908 | 12.01 | 0.916 | 0.834 | 14.21 | 0.913 | 0.773 | 14.92 | 0.906 | 0.757 | 15.04 | 0.889 | 0.750 |

### 4.2. Roles of Hidden Neurons

Figure 5a presents the connecting weights $w_{Ol,n}$ and bias $B_O$ of the output-layer neuron. By comparison with the bias, the most influential hidden neurons can be found. As can be seen, the bias is −0.38 and the weights for the hidden neurons $H_1$, $H_2$, $H_3$, and $H_8$ are 2.83, 0.26, 3.02, and −0.17, respectively. The positive neurons (e.g., $H_1$) would also increase (or decrease) the storm surge deviation in the next hour if output is positive (or negative). The negative neurons (e.g., $H_8$) would have an opposite impact. As for the rest of the neurons, their weights are less than 10% of the bias.

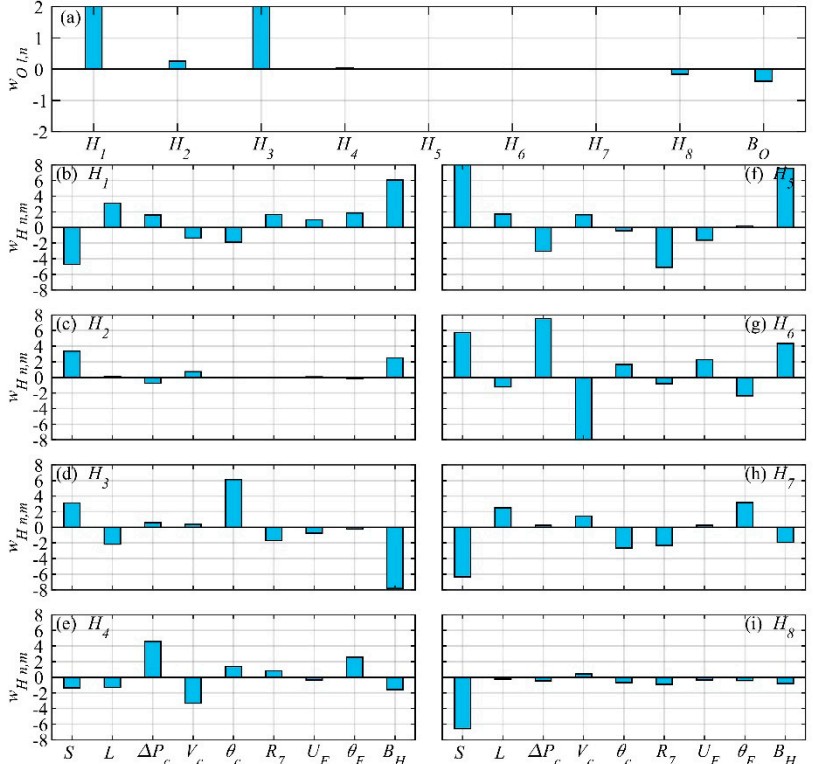

**Figure 5.** (**a**) Weights and bias of output-layer neuron ($w_{Ol,n}$ and $B_O$); (**b**–**i**) Weights and bias of each hidden neuron ($w_{Hl,n}$ and $B_H$).

Further, the weights $W_{H(n,m)}$ linking various input factors and the bias $B_H$ for each hidden neuron are shown in Figure 5b–i. The hidden neuron $H_1$ has a bias of 6.07 and weights of −4.73, 1.58, −1.36, −1.91, 3.10, and 1.65 (about 78%, 26%, 22%, 31%, 51%, and 27% of the bias) for the storm surge deviation ($S$), relative distance ($L$), typhoon central pressure deficit ($\Delta P_c$), maximum wind speed ($V_c$), relative angle ($\theta_R$), and radius of the typhoon ($R_7$), respectively. The hidden neuron $H_2$ presents a bias of 2.51 and receives corresponding inputs by weights of 3.36, −0.76, 0.72, −0.05, 0.10, and −0.02 (about 133%, 30%, 29%, 2%, 4%, 1% of the bias). The bias of hidden neuron $H_3$ is −7.83 while weights for the effective controlling factors are 3.10, 0.60, 0.39, 6.11, −2.17, and −1.69 (about 40%, 8%, 5%, 78%, 28%, and 22% of the bias). For the hidden neuron 8, the bias is −0.83 and weights are −6.58, −0.46, 0.42, −0.67, −0.22 and −0.88 (about 793%, 55%, 51%, 81%, 27%, and 106% of the bias).

We find that the hidden neurons $H_2$ and $H_8$ almost fully indicate the previous status of storm surge deviation. The hidden neurons $H_1$ and $H_3$ are both strongly biased, tending to output positive and negative unity (i.e., +1 and −1) through the activation function, respectively. If the previous storm surge level and relative angle of a typhoon could somewhat balance these biases, other controlling parameters would be able to affect the outputs of hidden neurons (e.g., see the weights of maximum wind speed, central pressure deficit, relative distance, and radius of typhoon in the hidden neuron $H_1$). With a smaller bias, the hidden neuron $H_2$ would more easily reflect the response of low surge level caused by

the variation of inputs (e.g., central pressure deficit). Interestingly, unlike the conventional regression analysis that seeks a single (positive) coefficient to relate the storm surge deviation to each forcing factor (e.g., $S = a \, \Delta P_c + b \, V_c^2 \, \cos\theta_w$), the artificial neural network with a number of hidden neurons utilizes several positive and negative weights for atmospheric forcing (e.g., see Figure 5b,d for the weights of maximum wind speed). Particularly, notice that a forcing factor (e.g., wind speed) varying from its lower to upper limit (i.e., a normalized input ranging from −1 to 1) might cause a different degree of influence. Such a nonlinear system would be better depicted by these negative/positive weights.

### 4.3. Contributions of Controlling Parameters

The contributions of controlling parameters to surge prediction are further revealed through examining the forward computation process at specified conditions, while their mixed and complicated contribution cannot be explicitly expressed. For example, if a typhoon (e.g., Krosa) with low pressure and high wind (e.g., $\Delta P_c = 88$ hpa; $V_c = 51$ m/s; $R_7 = 300$ km) is at some distance southeast of Taiwan (e.g., $L = 620$ km; $\theta_R = 304°$), no significant storm surge is generated either in the current (e.g., storm surge at the tidal station $S = 7$ cm) or next hour. The normalized inputs of storm surge deviation, central pressure deficit, wind speed, radius of typhoon, relative distance and angle are approximately −0.68, 0.87, 0.89, 1.0, 0.55, and 0.81, respectively. After computing the weighted sums and transformations, the most influential hidden neurons $H_1$, $H_2$, $H_3$, and $H_8$ give 1.0, 0.21, −1.0, and 0.98, respectively. The final result of output neuron is −0.67, yielding tiny surge variation for the next hour (e.g., $S = 7.6$ cm at the tidal station).

Subsequently, the typhoon maintains its status, keeps traveling toward Taiwan (e.g., $L = 111$ km $< R_7$; $\theta_R = 316°$), and affects coastal waters near the tidal station (e.g., surge $S$ with a dramatic increase from 97 to 169 cm in an hour). The wind speed and central pressure of the typhoon start to play some role in the hidden neuron $H_3$ while the current status of storm surge dominates the behaviors of hidden neurons $H_2$ and $H_8$. For this situation, the normalized inputs of surge deviation, central pressure deficit, wind speed, radius of typhoon, relative distance and angle are about 0.18, 0.87, 0.89, 1.0, −0.74, and 0.89, respectively. The major hidden neurons yield apparent changes in the intermediate outputs (i.e., $H_2 = 0.99$, $H_3 = −0.69$, and $H_8 = −0.99$), except for the hidden neuron $H_1$ (i.e., $H_1 \sim 0.99$). The final result of output-layer neuron is 0.86.

When the typhoon gets really close to the coastal area (or even makes landfall), in addition to its intensity the relative angle which determines local wind direction is a critical factor for surge variation (e.g., a decrease of $S$ from 164 to 128 cm). For instance, a typhoon with the same status moves to the south slightly away from the tidal station (e.g., $L = 55$ km; $\theta_R = 268°$). Accordingly, the normalized inputs of surge deviation, central pressure deficit, wind speed, radius of typhoon, relative distance and angle are about 0.82, 0.87, 0.89, 1.0, −0.88, and 0.60, respectively. The intermediate outputs of hidden neurons $H_1$, $H_2$, $H_3$, and $H_8$ are 0.68, 1.0, −0.52, and −1.0, respectively. A drop in $H_1$ due to the influence of relative location yields an obvious decrease in the final output, i.e., 0.48. In general, the roles of controlling parameters (and hidden neurons) are clarified. For other situations (e.g., a lower maximum surge generated by Typhoon Haitang with the same intensity but at a greater distance away from the coast), their contributions can be simply analyzed using the procedure above.

### 4.4. Knowledge of the Neural Network

We further extract knowledge from the artificial neural network (i.e., maximum storm surges under various conditions) using a series of synthetic typhoons. Based upon the historical data, the central pressure deficit considered in this study ranges from 43 to 88 hpa. The maximum wind speed from 32 to 50 m/s is determined by an empirical regression $V = 2.692 \, (\Delta P)^{0.654}$. The radius of typhoon is 250 km for all cases. The initial location is at a distance of 648 km southeastern of Longdong with a relative angle from 292° to 325°. Then, the typhoon moves along a straight line with a forward speed of 18 km/h when approaching Taiwan (see Figure 6a for the paths). After a period of 36 h, the typhoon finally makes landfall in Ilan, which is located approximately 15 km south of the station.

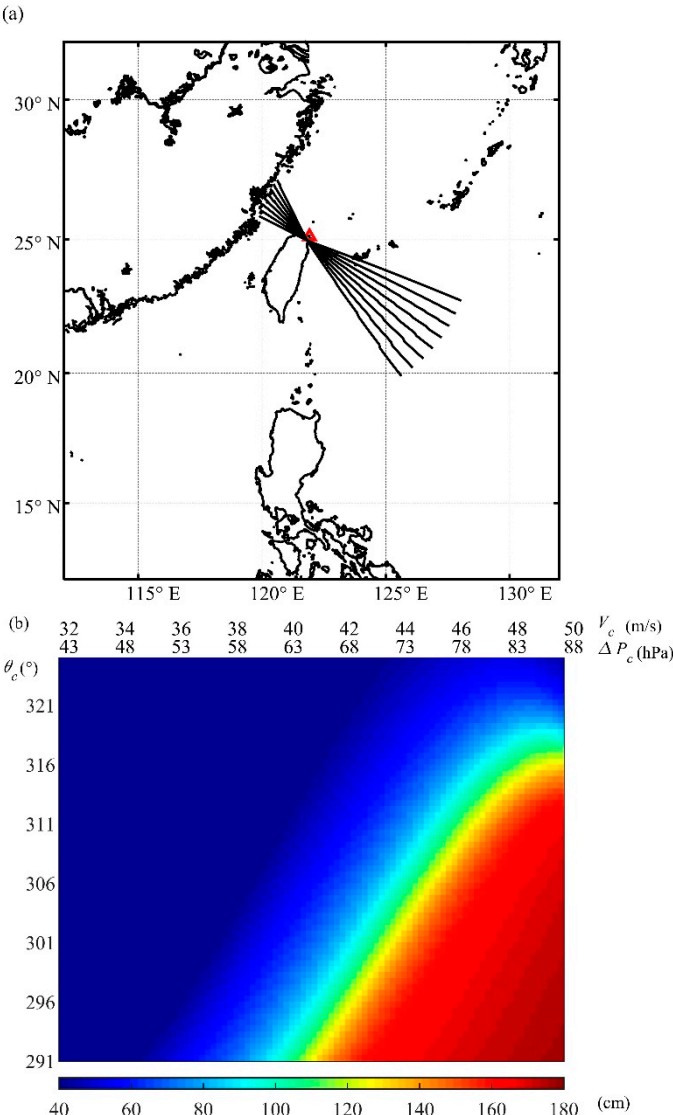

**Figure 6.** (**a**) Tracks of synthetic typhoon events; (**b**) predicted pattern of maximum storm surge under various scenarios.

Figure 6b presents the distribution of predicted maximum storm surge in terms of central pressure deficit, maximum wind speed, and relative angle. If the initial relative angle of the typhoon falls between 318° and 320°, the storm surge induced mainly by the pressure (i.e., the inverted barometer effect) has a maximum of 42 to 86 cm. On the other hand, the typhoon with an initial relative angle between 292° to 317° would provide a larger onshore wind component. Especially for the case of 292°, the maximum storm surge with a dramatic increase up to 180 cm can be generated by the local high wind and low pressure. Overall, the result is quite consistent with the actual observation data (during Typhoon Krosa) as well as the estimation from empirical equations [20,21] and numerical simulations [76,77]. The result of maximum storm surge also indicates an interesting fact that the inverted barometer effect (due to pressure deficit) and wind setup (due to wind speed) roughly are in the same order of magnitude for Taiwan coastal waters, unlike some other global cases where the low pressure makes a minimal contribution (about 5% of total) in comparison with the wind-driven surge (e.g., [8]).

*4.5. Long-Lead-Time Prediction*

Long-lead-time (from t + 3 to t + 12) storm surge predictions and the corresponding performance assessment during Typhoon Krosa are shown in Figures 7 and 8. Results for all the training events are presented using scatter plots in Figure 9. Note that another four hidden neurons were added using the constructive method [78] to achieve accurate 12-h-ahead prediction. Additionally, six combinations of inputs were considered to examine the influences of typhoon parameters on storm surge prediction. In particular, comparisons with the simple auto-correlation case (i.e., the storm surge at previous hour as the single input) are adopted to clearly demonstrate the marginal improvement among different cases.

The ANN model (S) fails to predict surge variation with obvious underestimations as the lead time increases (see the purple lines in Figure 7a and the star symbols in Figure 8a). For the three-h-ahead predictions, the statistical indices *RMSE*, *CC*, and *CE* (see Table 3) are 24.62 cm, 0.804, and 0.625, respectively. Subsequently, several static typhoon characteristics are included in the ANN model. Based on the information of relative distance, central pressure deficit, and maximum wind speed (together with relative angle), the ANN model (SLPV) is able to capture most of the surge pattern (see the dark yellow lines in Figure 7a and the star symbols in Figure 8d). Even in t+6 prediction, the performance assessment gives *RMSE* = 12.71 cm, *CC* = 0.965, and *CE* = 0.902. The predictions would be less satisfactory if the ANN model simply adopts the pressure without taking wind effect into account (i.e., SLP) or vice versa (i.e., SLV). Through the comparison between SLP and SLV (see the blue and green lines in Figure 7a), wind effect appears to be the key for accurate prediction. Additionally, the radius of the typhoon which quantifies the storm-affected area is employed in the inputs. Thus, the ANN surge model (SLPVR) increases its capability for 6-h-ahead prediction (not shown in Figure 7), i.e., *RMSE* = 10.49 cm, *CC* = 0.968 and *CE* = 0.933. Finally, dynamic factors of the typhoon are fully considered, i.e., forward speed and angle. In six-h-ahead predictions, the ANN model (SLPVRF) successfully predicts the peak surge (see the red lines in Figure 7b) caused by Krosa (*RMSE* = 10.01 cm, *CC* = 0.969, and *CE* = 0.939). Reasonable accuracy can still be maintained as the lead time increases up to 12 h, i.e., *RMSE* = 14.95 cm, *CC* = 0.938, and *CE* = 0.872. In general, the overall prediction skills of ANN model reduce with the lead time, e.g., the rapidly-accumulated *RMSE* of ANN model (S). However, prediction results are greatly improved by full consideration of typhoon parameters (including static and dynamic factors). In 12-h-ahead predictions, the ANN model (SLPVRF) yields the optimal improvement ratios (performance difference divided by the result from ANN model (*S*)) over 0.64, 0.87, 28 for *RMSE*, *CC*, and *CE*, respectively (see Figure 9).

Long-lead-time prediction results for validation event(s) are shown in Figures 10–12. Unlike the pattern with rapid growth and extremely high peak in Krosa, gradual surge variation during Typhoon Haitang is reasonably described by the three-h-ahead predictions using the ANN model (S). The statistics for prediction performance are *RMSE* = 16.92 cm, *CC* = 0.853, and *CE* = 0.670 (see Table 4). As the lead time extends, e.g., up to 12 h, the ANN model (S) demonstrates its poor skill, i.e., *RMSE* = 28.50 cm, *CC* = 0.472, and *CE* = 0.147. Similarly, long-lead-time predictions can capture overall variation of storm surge in validation event(s) fairly well while effective controlling parameters are fully considered in the ANN model (SLPVRF). The statistical indices *RMSE*, *CC*, and *CE* for 12-h ahead predictions are improved to 15.04 cm, 0.889, and 0.750, respectively. The corresponding improvement ratios are 0.47, 0.86, and 4.1.

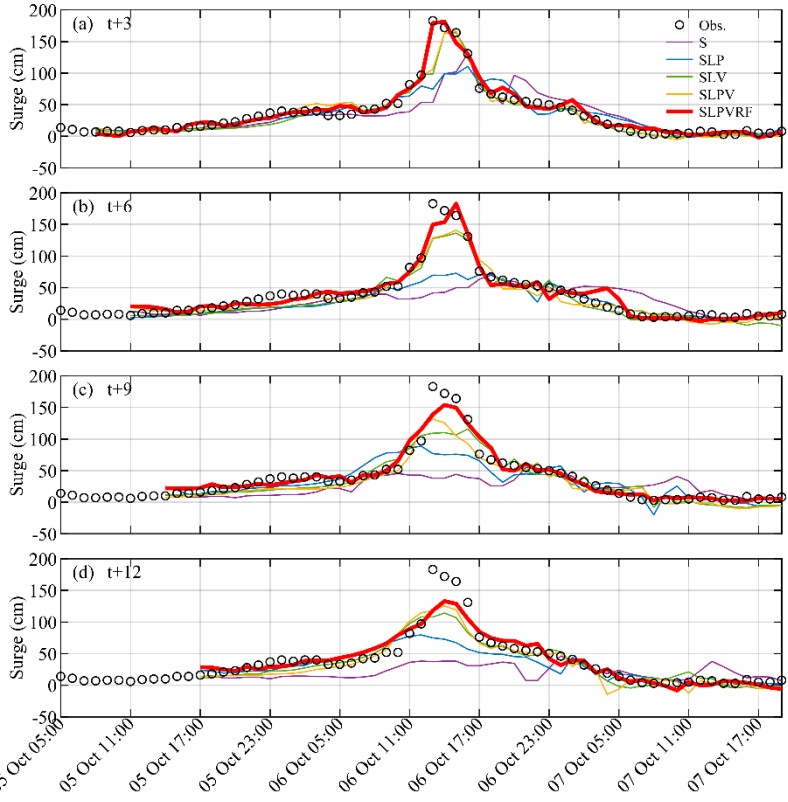

**Figure 7.** Long-lead-time storm surge predictions obtained by different combinations of controlling parameters for Krosa event (training): (**a**) t + 3; (**b**) t + 6; (**c**) t + 9; (**d**) t + 12.

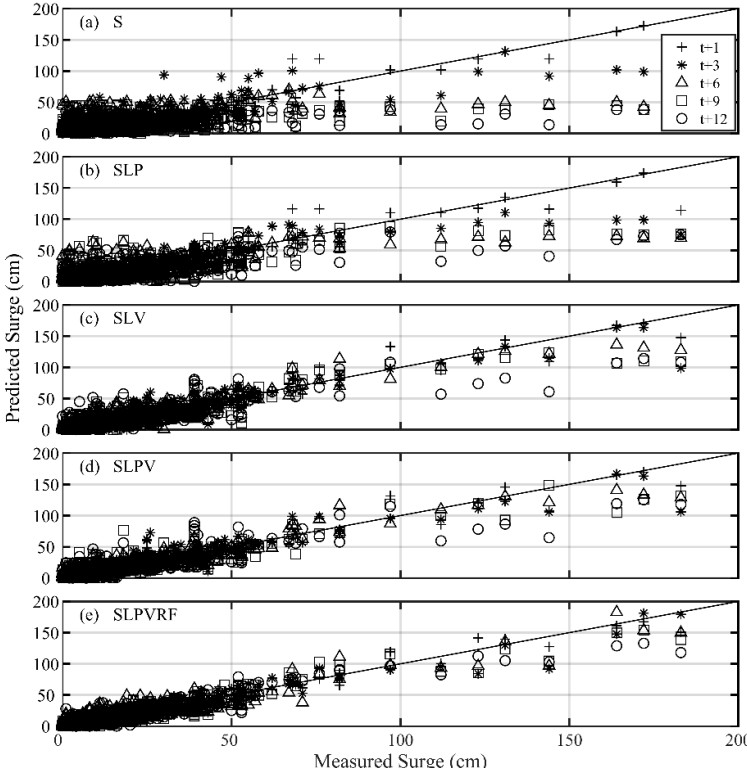

**Figure 8.** Scatter plots of predicted and measured storm surges for all training events under different input combinations (from top to bottom panels) and lead-time (various symbols): (**a**) *S*, (**b**) SLP, (**c**) SLV, (**d**) SLPV, and (**e**) SLPVRF.

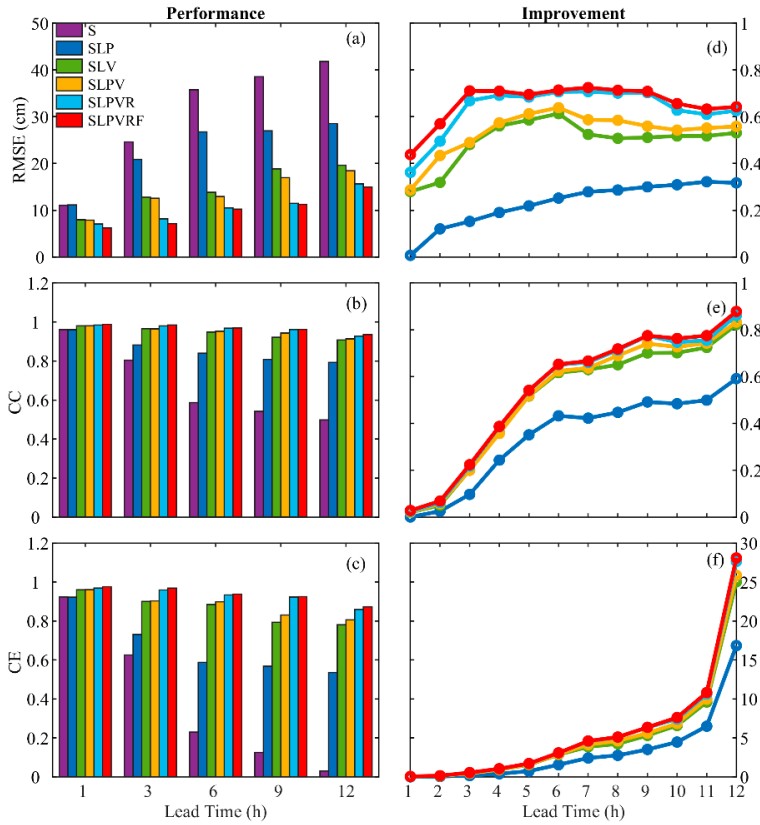

**Figure 9.** Prediction capability and performance improvement in terms of (**a**,**d**) root mean square error (*RMSE*); (**b**,**e**) correlation coefficient (*CC*); and (**c**,**f**) coefficient efficiency (*CE)* for all the training events.

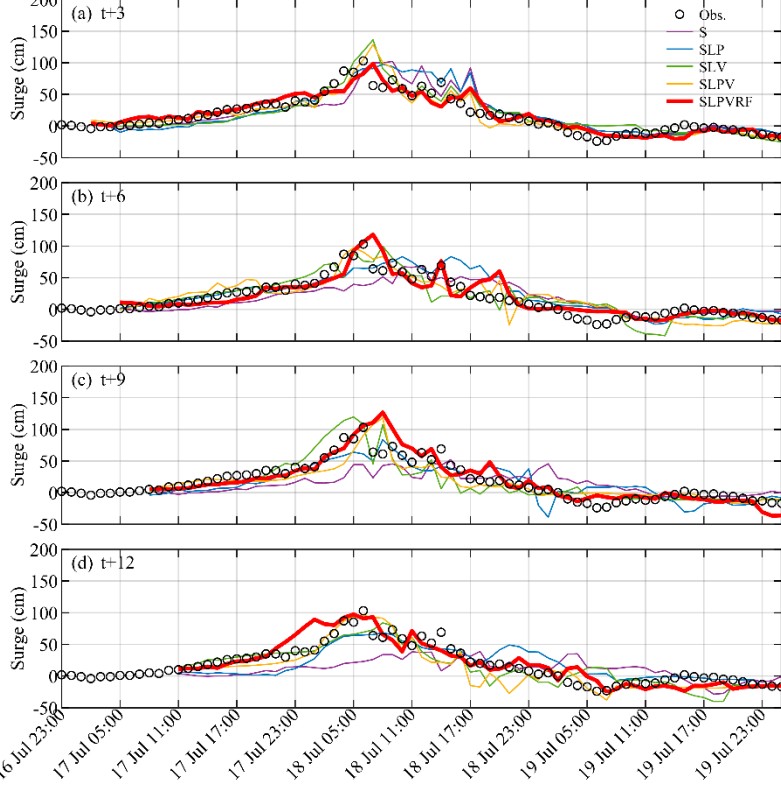

**Figure 10.** Long-lead-time storm surge predictions obtained by different combinations of controlling parameters for Haitang event (validation): (**a**) t + 3; (**b**) t + 6; (**c**) t + 9; (**d**) t + 12.

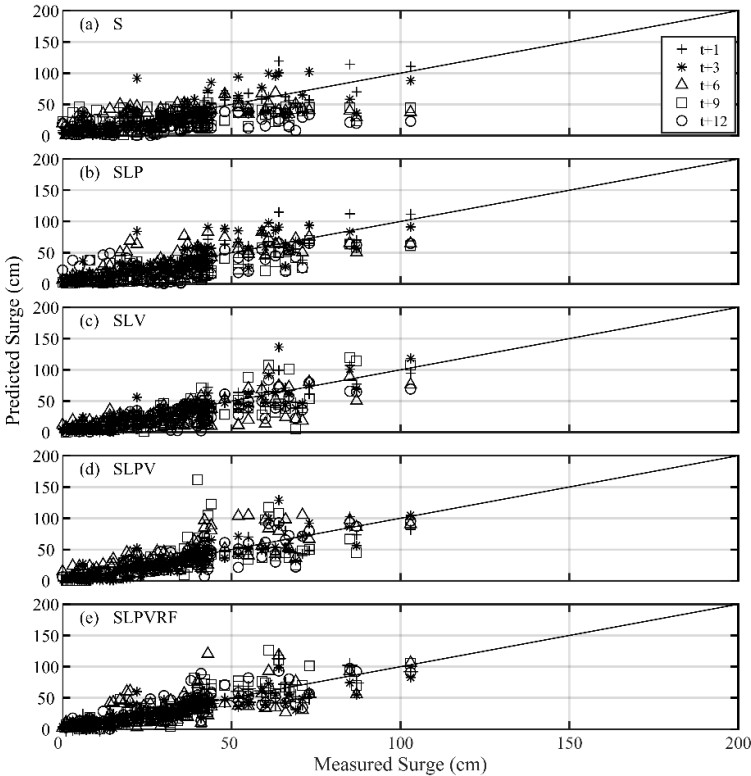

**Figure 11.** Scatter plots of predicted and measured storm surges for all validation events under different input combinations (from top to bottom panels) and lead-time (various symbols): (**a**) *S*, (**b**) SLP, (**c**) SLV, (**d**) SLPV, and (**e**) SLPVRF.

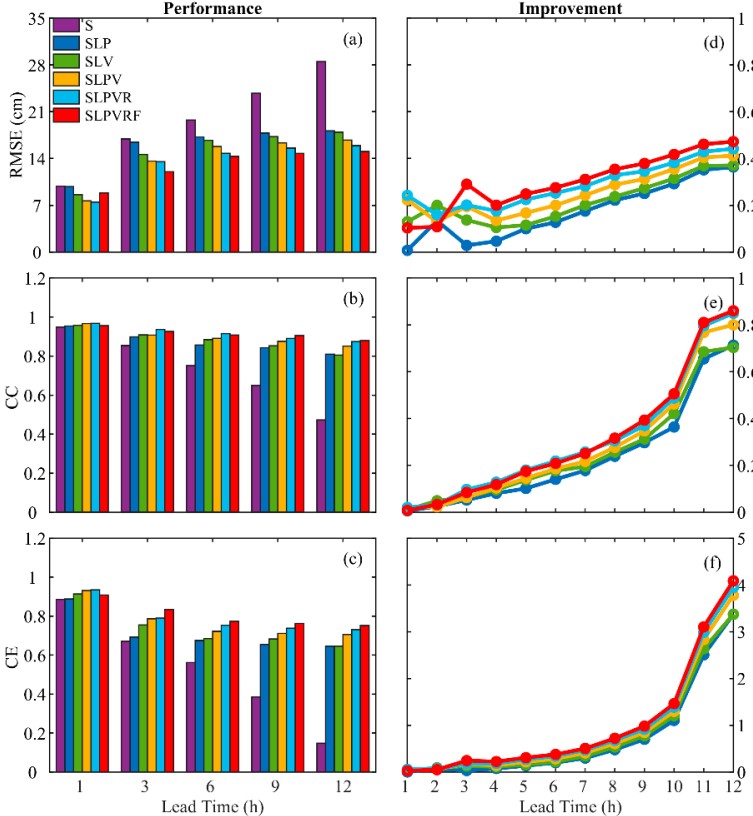

**Figure 12.** Prediction capability and performance improvement in terms of (**a**,**d**) *RMSE*; (**b**,**e**) *CC*; and (**c**,**f**) *CE* for all the validation events.

Over the past decades, attempts that utilize artificial neural networks to efficiently predict storm surge for Taiwan coastal waters have been made [16,21,62]. However, the prediction lead time was quite limited, e.g., only one hour [16]. Tseng et al. [62] reported acceptable results for three-h ahead prediction using an artificial neural network surge model with consideration of 18 input parameters (including local meteorological information and characteristics of typhoon). In this study, reasonable prediction accuracy is successfully extended to a much longer lead time, i.e., 12 h. Possible reasons for further improvement between the previous and present studies are discussed as follows.

First, local meteorological information (e.g., low wind) would not imply potential future influence since surge is mainly induced by high wind and low storm pressure. Besides, local meteorological information would (almost) become identical to the characteristics of the typhoon when a typhoon moves very close to the study area. Hence, local meteorological information seems to be redundant, in comparison with the dominant typhoon factors which can reflect longer temporal variation of storm surge. Particularly, more (redundant) factors considered in the neural network would not guarantee better results as the high dimension of inputs might complicate the learning (training) processes with unwanted issues of local optimization or over-fitting.

Second, to characterize the dynamic/kinematic condition of a typhoon, an invasion angle defined as the difference between the relative and moving directions of a storm was utilized in the earlier study (see Figure 2 in [62]). By this definition, one would always obtain a value of zero for this angle if a typhoon from any location directly heads toward the tidal station. Thus, the influences caused by different situations above could not be properly indicated using the invasion angle. In this paper, with proper examination and selection of essential input parameters, the ANN model (SLPVRF) successfully extends reasonable prediction accuracy to a much longer lead time (i.e., $CE > 0.7$ for t + 12). A dedicated ANN model for different locations along Taiwan coastline can be easily built in a similar way. Based upon the operational advantages (i.e., fast computations) together with good capability for long-lead-time prediction, potentially the newly-developed ANN model can be used in the near future for coastal disaster preparedness and early warning.

## 5. Conclusions

Storm surge induced by severe typhoons has caused many catastrophic tragedies to coastal communities over the past decades [3,4]. To achieve coastal disaster mitigation, accurate and efficient prediction (assessment) of storm surge is still an important task [8–13]. In this paper, we revisit the topic of storm surge prediction using artificial neural networks and effective typhoon parameters. The progress of storm surge modeling with different types of methodologies (i.e., empirical formulas, hydrodynamic models, and artificial intelligence approaches [2,15,16]) is reviewed and the remaining unresolved issues are addressed. In the present study, therefore, we develop a new ANN-based storm surge prediction model with the main objectives of extending further applicability and gaining deeper insight, i.e., the crucial prediction lead-time and useful information inside the black box for disaster management.

In this paper, we choose the northeastern region of Taiwan as the study area, where the largest storm surge record (over 1.8 m) has been observed during Typhoon Krosa. After data collection (from 2005 to 2014), by analogy with the physical modeling approach, key factors for describing the impact of a typhoon are carefully selected to develop the ANN-based storm surge model (for various lead-times from 1 to 12 h). The values of effective typhoon parameters are fully based on real observation rather than numerical weather predictions to avoid common issues of atmospheric models (i.e., accuracy and uncertainty). A knowledge extraction method (KEM) based upon backward tracking and forward exploration procedures is also proposed to analyze the roles of hidden neurons and controlling parameters in storm surge prediction as well as to reveal abundant useful information from the fully-trained artificial brain. Last, the capability of the ANN model for long-lead-time predictions and influences of controlling parameters are investigated. Five major findings are summarized as follows:

1. One-hour-ahead predictions by the newly-developed ANN model capture both peak surge and rising/falling pattern accurately in both training and validation phases (see Tables 3 and 4 for statistical indices of performance). Input parameters used in the model include storm surge itself and effective typhoon parameters, i.e., the central pressure deficit, maximum wind speed, relative location (described by distance and angle), radius of the typhoon, and forward speed and direction. Note that similar performances for short-term storm surge prediction (see Table 3) can be achieved either by using the ANN model without a complete consideration of typhoon parameters, or even by taking a simple extrapolation method. The objective of a reliable one-h-ahead prediction model is not for early warning but to facilitate further analysis of the roles/contributions of hidden neurons and typhoon parameters in this study.

2. In our current network structure, the backward tracking procedure of KEM finds the four most influential hidden neurons, i.e., $H_1$, $H_2$, $H_3$, and $H_8$. The positive neurons would increase (or decrease) the storm surge deviation in the next hour if output is positive (or negative). The negative neurons would give an opposite impact. Besides, the hidden neurons $H_2$ and $H_8$ mainly indicate the previous status of storm surge deviation. For the hidden neurons $H_1$ and $H_3$, typhoon parameters are able to alter the outputs while the strong biases are balanced by storm surge level and relative angle of the typhoon.

3. The wind speed and central pressure deficit play some role in the hidden neuron $H_3$ for storm surge prediction (e.g., surge *S* with a dramatic increase from 97 to 169 cm in an hour) while a typhoon travels toward Taiwan and affects coastal waters near the tidal station. When the typhoon is really close or even makes landfall, in addition to its intensity the relative angle which determines local wind direction is a critical factor for surge variation. In our case study, the relative angle causes a drop in $H_1$, yielding an obvious decrease in the final output (e.g., a decrease of *S* from 164 to 128 cm).

4. The forward exploration procedure of KEM successfully reveals a general pattern of maximum storm surge under various scenarios (e.g., the max S induced by a typhoon with relative angel of 292° increases over 180 cm). Interestingly, the inverted barometer effect (due to pressure deficit) and wind setup (due to wind speed) roughly are in the same order of magnitude for Taiwan coastal waters, unlike some other global where the low pressure makes a minor contribution (e.g., 5% of total) in comparison with the wind-driven surge [8].

5. Satisfactory accuracy for a much longer lead time is achieved, e.g., training with *CE* > 0.85 and validation with *CE* > 0.75 in 12-h-ahead predictions. Six combinations of inputs clearly demonstrate the influences of typhoon parameters on surge prediction. Reasons for further improvement over earlier works [16,60] are also given. In previous studies, inputs for local meteorological conditions would lead to higher dimensions but fail to indicate potential influences of a typhoon, hindering the learning and prediction capability of ANN-based surge models. Besides, for the cases where a typhoon from any direction exactly passes through the study area, the invasion angle would always be zero and cannot correctly reflect the resulting influences.

Overall, the analyses in this study try to give an answer to enduring questions (lead time and inner knowledge) in ANN-based storm surge modeling. The success of the newly-developed ANN model with effective typhoon factors is proved/supported by satisfactory long-lead-time predictions. With great operational advantages (i.e., both of accuracy and efficiency for storm surge computation), potentially the ANN model can be further utilized in coastal disaster management.

**Author Contributions:** Conceptualization, W.-T.C., C.-C.Y., T.-W.H., and W.-C.L.; methodology, W.-T.C., C.-C.Y., and W.-C.L.; validation, W.-T.C. and C.-C.Y.; formal analysis, W.-T.C., C.-C.Y., and C.-Y.L.; investigation, W.-T.C., C.-C.Y., and C.-Y.L.; resources, T.-W.H.; writing—original draft preparation, W.-T.C. and C.-C.Y.; writing—review and editing, W.-T.C. and C.-C.Y.; visualization, W.-T.C. and C.-C.Y.; supervision, C.-C.Y. and T.-W.H.; project administration, C.-C.Y. and T.-W.H.; funding acquisition, T.-W.H. All authors have read and agreed to the published version of the manuscript.

**Funding:** This research was funded by Ministry of Science and Technology, Taiwan grant number 106-2221-E-019-074-, 107-2221-E-019-010-MY3 and 108-2218-E-019-001-MY2.

**Acknowledgments:** The authors thank the anonymous reviewers for their insightful and constructive comments to improve this paper. This study was provided by the Ministry of Science and Technology (MOST) and Ministry Education, Taiwan. The observation data used in this study were measured and provided by the Central Weather Bureau of Taiwan. The authors would like to express great thanks for all the supports.

**Conflicts of Interest:** The authors declare no conflict of interest.

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
