# Peer review of "Long-Lead-Time Prediction of Storm Surge Using Artificial Neural Networks and Effective Typhoon Parameters: Revisit and Deeper Insight"

_water, doi:10.3390/w12092394_

Round 1

Reviewer 1 Report

Some statements of the manuscript should be corrected and/or explainedQ

lines 197-199: "The total water level ... coastal waters". Should be documented.

line 211: "e.g. ..." Is not clear.

line 225: the note 1 is not in the main text.

lines 425-426: the order of plots 7, 8 and 9 do not correspond to the presentation of plotsin pages 15 and 16.

In this work a storm surge prediction for a time of 12 hours is proposed. This is not a long-lead-time prediction. Therefore the authors shoould comment on that i.e. to support in more details why their prediction method is a "long-lead-time" prediction.  

Author Response

Response to Reviewer 1:

Point 1: Some statements of the manuscript should be corrected and/or explained.

Response 1: We appreciate the suggestions from reviewer 1 for further improvement of this paper. We totally agree with the comments and have revised the manuscript accordingly. The response for each comment is given as follows.

Point 2: Lines 197-199: "The total water level ... coastal waters". Should be documented.

Response 2: The characteristics of coastal water level has been studied by several researchers in Taiwan. For example, Chiou et al. [72] pointed out that the interaction between surges and tides is insignificant especially for the northeastern coastal waters of Taiwan. Please see lines 202-205 in the revised manuscript (marked in red color).

Lines 202-205:

The total water level can be obtained later simply by a linear superposition since nonlinear effects between surges and tides are mild (e.g., about 10% increase in total water level along the west coast) or even insignificant especially for the northeastern coastal waters of Taiwan [72].

Point 3: line 211: "e.g. ..." Is not clear.

Response 3: The original sentences have been revised for better understanding. See lines 214-219 for the revision (in red color).

Lines 214-219:

In addition to storm surge (S), central pressure deficit (∆Pc), and maximum wind speed (Vc), we consider the location of a typhoon by its distance (L) and relative angle c) rather than the longitude and latitude. Particularly, the relative angle of a typhoon (measured counterclockwise with 0 degree from the east) can imply the favorable wind direction (e.g., θw = θc +115°) near a tidal station which is an important factor for storm surge prediction. Note that only the relative angle is needed here since a linear shift between θw and θc causes no difference for the artificial neural network model.

Point 4: line 225: the note 1 is not in the main text.

Response 4: The note 1 for Table 2 has been removed.

Point 5: lines 425-426: the order of plots 7, 8 and 9 do not correspond to the presentation of plots in pages 15 and 16.

Response 5: In section 4.5 (i.e., pages 15-16), more detailed information for the corresponding plots has been provided in our discussions.

Line 447: The ANN model (S) fails to predict surge variation with obvious underestimations as the lead time increases (see the purple lines in Figure 7a and the star symbols in Figure 8a).

Line 452: the ANN model (SLPV) is able to capture most of the surge pattern (see the dark yellow lines in Figure 7a and the star symbols in Figure 8d)

Line 456: Through the comparison between SLP and SLV (see the blue and green lines in Figure 7a), wind effect appears to be the key for accurate prediction.

Line 461: In 6-h-ahead predictions, the ANN model (SLPVRF) successfully predicts the peak surge (see the red lines in Figure 7b) caused by Krosa (RMSE = 10.01 cm, CC = 0.969, and CE = 0.939).

Line 467: In 12-h-ahead predictions, ANN model (SLPVRF) yields the optimal improvement ratios (performance difference divided by the result from ANN model (S)) over 0.64, 0.87, 28 for RMSE, CC, and CE, respectively (see Figure 9).

Point 6: In this work a storm surge prediction for a time of 12 hours is proposed. This is not a long-lead-time prediction. Therefore, the authors should comment on that i.e. to support in more details why their prediction method is a "long-lead-time" prediction.  

Response 6: After literature review, we found that earlier studies carried out storm surge prediction mostly with the lead time of 1 hour. In the past decade, 3-hour ahead storm surge prediction was achieved using a recursive approach [62]. Yet, the prediction skills were still quite limited in terms of lead time and accuracy. Recently, Kim et al. [10] showed successful predictions of after-runner storm surges in a much longer lead-time (e.g., 5, 12, 24 hours). In this study, with a similar definition (i.e., a lead time up to 12 hours), we therefore proposed long-lead-time storm surge prediction to meet the demand in practical decision-making process. More details have been provided in the revised manuscript.

Lines 124-136:

In the former problem, given predictions with an insufficient lead-time (e.g., 1-hour in most previous works), effective coastal disaster management (prevention/mitigation) through preparedness efforts and early warning systems would be hardly achieved. In the past decade, 3-hour ahead storm surge predictions with a recursive approach were attempted [62], yet the prediction skills in terms of lead time and accuracy were still quite limited. Recognized the urgent demand in practical decision-making process, Kim et al. [10] recently re-examined ANN approaches and showed successful predictions of after-runner storm surges on Tottori coast, Japan in a much longer lead-time (e.g., 5, 12, 24 hours). In their work, the ANN models were appropriately trained and tested to reproduce three representative (i.e., 100-year return period) surge events induced by severe typhoons (e.g., typhoon Maemi with a minimum central pressure of 910 hPa) moving in a similar track. For the areas with frequent typhoon invasions from variable paths (e.g., 10 major paths in Taiwan), however, accurate long-lead-time (defined by a lead time up to 12 hours in this study) surge prediction is still challenging.

Reviewer 2 Report

This is an interesting and relevant paper. A few points may need some explanation and at some places things should be more precise.

It might be good to explain around line 53 that surge predictions can be used in two ways: (1) real time predictions in order to warn people mitigate the effects of the coming surge in time and (2) statistical prediction to determine what the “once in xxx-years” surge will be in order to build protective structures, etc. For both purposes different types of models have to be used. For both purposes an ANN-approach is possible, but the described ANN approach in this paper is focused on the first purpose. I think it might be good for the reader to make this point clear.

Line 58: quite bluntly is given here that a pressure drop of 1 mb leads to a 1-cm rise in sea level. This is usually a negligible value compared to the surge itself. But, because of the shape of the coast, such a gustbump may lead to a seiche, which due to resonance may cause considerable rise in the surge level. Physically this is complicated to compute, but in an ANN (for a given fixed location) this effect is automatically included. This is an additional advantage of an ANN.

Line 66/63: Maybe it is good to mention that a pure statistical approach to only surge-observations works quite well for a given location. In fact this is the oldest approach. Already in 1938 Wemelsfelder in the Netherlands found a log-relation between the value of the surge level and the probability of occurrence (report is only available in Dutch: http://resolver.tudelft.nl/uuid:0840f657-17bc-4274-a8a4-49f5c1749e83). After the 1953 flood this was elaborated by prof. van Dantzig as a basis for prediction of the extreme design surge, see for example his paper for the Int. congress of Mathematicians (https://www.mathunion.org/fileadmin/ICM/Proceedings/ICM1954.1/ICM1954.1.ocr.pdf ) in 1954 (on  p 218). This paper is in English.

line96: it is suggested that ANN is useful for storm surge and for tide prediction.  For tides this is doubtful, because an astronomical tide can be predicted rather exact using the appropriate coefficients.

Line 104: An ANN is only the first step of human learning. First humans start to find relations, and then they start to explain the relation in physical terms, in order to be able to use the relations also outside the range of observed data. In principle an ANN is not doing the last step. But you tried in your paper to analyze the role of the hidden neurons. In fact this is typically a step in the second phase of human learning. In fact you stress this in line 135. But this cast might be a more structured in your paper.

Line 172: in the table you don’t mention the landfall location and the path of the typhoon. Later you came back to this and use these elements as elements in the ANN. Maybe it is good to mention that already at this point.

When looking to figure 2 I wondered if the orientation of the coast is relevant. In fact it is, but because the ANN is developed for a specific point, the orientation of the coast is implicitly included in the model. This also means that the model works only for this specific location. Maybe you can explain this at this point.

Line 299 and further: The one hour ahead prediction gives quite good results, but I think that even a very stupid prediction (take the value of the last two hours and extrapolate linearly with one hour) will give a result with the same quality. So in fact the conclusion of this section is that an ANN for one-hour prediction does not give any added value. In fact the importance of the one-hour prediction exercise is not to make a reliable model (because even stupid extrapolation works as well), but to analyze the role of the hidden neurons. This should be stressed somewhat more.

In fact the most important part is section 4.5 where you make long-lead-time predictions.

What I miss after line 500 is that a prediction model for the surge (in fact your ANN) depends fully on the quality of the prediction of the behaviour of the typhoon. In fact all your conclusions are only correct if we are able to predict the magnitude and path of the typhoon.

Line 504: you mention climate change. This is not a topic in you paper and also not relevant for this paper. The effect of climate change is on the occurrence and magnitude of the typhoon. But this model uses that info as input. So, although fashionable, I would omit the line about climate change.

Line 507: you mention that you thoroughly reviewed the development of storm surge prediction. In fact you mention some of the papers on storm surge prediction, but you did not critically review them. I also don’t think that this is useful in the framework of this paper but then you should not mention that you did a thorough review.

Lines 557-564 is in my opinion not a very useful paragraph. I think it is too vague and fits more in a social science paper than in a hard core engineering paper. I would not include this.

On the other hand I think you did not mention one of the very big operational advantages of a good ANN model with a long-lead-time prediction. The model is very fast. You can easily make a dedicated model for many relevant locations along the Taiwanese coast. And on routine basis all these models can be run every hour to give every hour an updated surge prediction for the next 12 hours. Making the red curve of fig 10d is can be done easily every hour. A hydrodynamic model can for the time being not do this on an hourly basis.

Author Response

Response to Reviewer 2:

Point 1: This is an interesting and relevant paper. A few points may need some explanation and at some places things should be more precise.

Response 1: We appreciate the suggestions from reviewer 2 for further improvement of this paper. We totally agree with the comments and have revised the manuscript accordingly. The response for each comment is given as follows.

Point 2: It might be good to explain around line 53 that surge predictions can be used in two ways: (1) real time predictions in order to warn people mitigate the effects of the coming surge in time and (2) statistical prediction to determine what the “once in xxx-years” surge will be in order to build protective structures, etc. For both purposes different types of models have to be used. For both purposes an ANN-approach is possible, but the described ANN approach in this paper is focused on the first purpose. I think it might be good for the reader to make this point clear.

Response 2: Applications of surge predictions in two different ways has been addressed in the revised manuscript (see Lines 50-53 marked in red). Statement of our main purpose, i.e., early warning, in this study is also given in the introduction.

Lines 50-53: In order to achieve coastal hazard mitigation (i.e., early warning for a coming event as well as appropriate designs of protective structures), continuous efforts have still been made to better predict storm surge in real time and statistically assess its extreme variation [8-13].

Lines 141-144: In this paper, therefore, we revisit the topic of storm surge prediction using artificial neural networks and effective typhoon parameters with the main purpose of extending further applicability (i.e., the prediction lead time for early warning) and gaining a deeper insight (i.e., knowledge inside the neural network).

Point 3: Line 58: quite bluntly is given here that a pressure drop of 1 mb leads to a 1-cm rise in sea level. This is usually a negligible value compared to the surge itself. But, because of the shape of the coast, such a gustbump may lead to a seiche, which due to resonance may cause considerable rise in the surge level. Physically this is complicated to compute, but in an ANN (for a given fixed location) this effect is automatically included. This is an additional advantage of an ANN.

Response 3: Thanks for the comments. Indeed, some small atmospheric disturbance would cause tiny increase in sea water level, which can be further amplified by coastal topography/geometry due to resonance. In this study, the pressure drop for a severe typhoon is quite large and able to cause notable rise in surge level. In order NOT to mislead the readers, we have corrected some parts of original statements in the introduction. The effects of coastal topography/geometry, i.e., shoaling and resonance, have also been addressed.

Lines 56-62: Storm surge induced by typhoons mainly consists of (i) wind setup (i.e., strong onshore wind causes significant rise of sea level) and (ii) pressure setup (e.g., a large pressure drop of 100 mb during a severe typhoon event leads to a 100-cm rise in sea level). In addition to the atmospheric forcing, physical processes in ocean motions (e.g., interactions with waves) and variation of coastal topography/geometry (e.g., influences of shoaling and resonance) would further complicate the magnitude of storm surge (e.g., a considerable rise in surge level) at a shallow-water study area.

Point 4: Line 66/63: Maybe it is good to mention that a pure statistical approach to only surge-observations works quite well for a given location. In fact, this is the oldest approach. Already in 1938 Wemelsfelder in the Netherlands found a log-relation between the value of the surge level and the probability of occurrence (report is only available in Dutch: http://resolver.tudelft.nl/uuid:0840f657-17bc-4274-a8a4-49f5c1749e83). After the 1953 flood this was elaborated by prof. van Dantzig as a basis for prediction of the extreme design surge, see for example his paper for the Int. congress of Mathematicians (https://www.mathunion.org/fileadmin/ICM/Proceedings/ICM1954.1/ICM1954.1.ocr. pdf ) in 1954 (on  p 218). This paper is in English.

Response 4: More details of statistical approach have been provided including good empirical relationships between maximum surge height and typhoon characteristics at a given location and log-relation for the prediction of extreme design surge. See Lines 66-71 in the revised manuscript.

Lines 66-71: Through statistical analysis (e.g., linear regression) of observation data, these earlier works obtained good agreement for the relationships between maximum surge height and typhoon characteristics (e.g., pressure deficit and wind speed) at some given locations. Typically, this simple tool can account for half of total variability of storm surge (with correlation coefficient CC ~ 0.6). Besides, a log-relation between the surge level and its probability of occurrence has been found and used for the prediction of extreme design surge [22].

Point 5: line96: it is suggested that ANN is useful for storm surge and for tide prediction.  For tides this is doubtful, because an astronomical tide can be predicted rather exact using the appropriate coefficients.

Response 5: Typically, the astronomical tide can be predicted really well using harmonic analysis with appropriate coefficients (amplitude and phase of each tidal constituent). In this study, we also utilized harmonic analysis to separate the non-tidal signal (i.e., storm surge) from the observation data. When the data length is not long enough, however, harmonic analysis may encounter some difficulty. Based on literature review, we found interesting and successful application of ANNs for tidal predictions. The reference [55] has been provided in the introduction.

Lines 98-99: The artificial intelligent (AI) approaches provide an alternative way for predicting storm surge (e.g., see [11,16,52-54]) as well as tidal variation [55].

  1. Lee, T.L. Back-propagation neural network for long-term tidal predictions. Ocean Eng. 2004, 31(2), 225-238.

Point 6: Line 104: An ANN is only the first step of human learning. First humans start to find relations, and then they start to explain the relation in physical terms, in order to be able to use the relations also outside the range of observed data. In principle an ANN is not doing the last step. But you tried in your paper to analyze the role of the hidden neurons. In fact, this is typically a step in the second phase of human learning. In fact, you stress this in line 135. But this cast might be a more structured in your paper.

Response 6: We thank the reviewer for his supportive comments. As mentioned, the roles of the hidden neurons and the contribution of controlling parameters in ANNs have been rarely touched and seem to be a difficult puzzle. In this study, we tried to and aimed at resolving such a long-lasting issue by the proposed knowledge extraction method. More structured and in-depth discussions have been made for the contributions of hidden neurons and effective typhoon parameters. In the near future, other coastal processes/phenomena and their controlling factors (e.g., geographical parameters) can be analyzed and discussed following a similar way.

Point 7: Line 172: in the table you don’t mention the landfall location and the path of the typhoon. Later you came back to this and use these elements as elements in the ANN. Maybe it is good to mention that already at this point.

Response 7: Thanks for the suggestions. We have provided further information about the items of typhoon data (i.e., location and intensity) in the revised manuscript. To predict storm surges, in fact, the input items are all the same no matter for historical or synthetic typhoon events in this study. For historical typhoons with known tracks, we directly read the locations (and calculated the relative distance and angle) as well as those parameters (e.g., pressure drop, wind speed, etc.). For the so-called synthetic typhoons, a simple track (a straight line commonly used for evaluating typhoon impacts [74]) was considered and determined by different relative angles and center location at the landfall. Some explanations have been given in the revised manuscript.

Line 170-174: Available information includes the track/path (i.e., center locations in latitude and longitude) and the intensity of each typhoon. Table 1 summarizes the central pressure (Pc), max. wind speed (Vc), and radius of typhoon (R7) for the selected events, i.e., Haitang (2005), Talim (2005), Longwang (2005), Bilis (2006), Kaemi (2006), Sepat (2007), Krosa (2007), Kalmegi (2008), Fungwong (2008), Sinlaku (2008), Morakot (2009), Soala (2012), and Matmo (2014).

Line 282-283: Note that the input items are all the same but their values (combinations) should be appropriately specified.

Line 286-288: For the path, a simple straight line commonly used for evaluating typhoon impacts is considered [74]. In other words, the tracks can be easily determined using different relative angles and center location at the landfall.

Point 8: When looking to figure 2 I wondered if the orientation of the coast is relevant. In fact, it is, but because the ANN is developed for a specific point, the orientation of the coast is implicitly included in the model. This also means that the model works only for this specific location. Maybe you can explain this at this point.

Response 8: The reviewer’s comment is true. Further explanation is given in the revision.

Line 234-237: The geographical parameter is not considered here. However, the effects of coastal topography/geometry on surge level at a specific site can be implicitly included in the artificial neural network model. For various tidal stations, a general (or universal) approach for ANN models will be further investigated in the future.

Point 9: Line 299 and further: The one hour ahead prediction gives quite good results, but I think that even a very stupid prediction (take the value of the last two hours and extrapolate linearly with one hour) will give a result with the same quality. So in fact the conclusion of this section is that an ANN for one-hour prediction does not give any added value. In fact the importance of the one-hour prediction exercise is not to make a reliable model (because even stupid extrapolation works as well), but to analyze the role of the hidden neurons. This should be stressed somewhat more.

Response 9: The reviewer correctly pointed out our objective. Indeed, the purpose of this example is not for early warning. The fully trained 1-h-ahead prediction model is further applied for analyzing the roles of hidden neurons and controlling parameters. The objective is clearly stressed in the beginning of section 4 and mentioned again in the conclusion in the revised manuscript.

Lines 304-307: Note that the objective of a reliable 1-h-ahead short-term prediction is not for early warning but to facilitate further analysis of the roles of hidden neurons and typhoon parameters using backward tracking procedure of KEM in Sections 4.2 and 4.3.

Lines 547-551: Note that quite similar performance for short-term storm surge prediction (see Table 3) can be achieved either using the ANN model without a complete consideration of typhoon parameters or even taking a simple extrapolation method. The objective of a reliable 1-h-ahead prediction model is not for early warning but to facilitate further analysis of the roles/contributions of hidden neurons and typhoon parameters in this study.

Point 10: In fact the most important part is section 4.5 where you make long-lead-time predictions.

Response 10: We appreciate the reviewer’s comments. A lot of efforts have been made to achieve satisfactory long-lead-time predictions in this study.

Point 11: What I miss after line 500 is that a prediction model for the surge (in fact your ANN) depends fully on the quality of the prediction of the behaviour of the typhoon. In fact all your conclusions are only correct if we are able to predict the magnitude and path of the typhoon.

Response 11: We agree with the reviewer’s comment if the inputs of ANN models are obtained from numerical weather predictions. As mentioned by the reviewer, accuracy and uncertainty of atmospheric models would cause another critical issue for storm surge prediction. In this study, therefore, the values of effective typhoon parameters are fully based on the real observation. In order NOT to mislead the readers, more details have been provided in our revised manuscript.

Lines 231-234: Note that the values of effective typhoon parameters are fully based on the real observation. Although they also can be obtained from numerical weather predictions, accuracy and uncertainty of atmospheric models would be another critical issue.

Lines 534-535: The values of effective typhoon parameters are fully based on real observation rather than numerical weather predictions to avoid common issues of atmospheric models (i.e., accuracy and uncertainty).

Point 12: Line 504: you mention climate change. This is not a topic in your paper and also not relevant for this paper. The effect of climate change is on the occurrence and magnitude of the typhoon. But this model uses that info as input. So, although fashionable, I would omit the line about climate change.

Response 12: We agree with the comment. The effect of climate change is not in the scope of this study. In conclusion, the part of climate change has been removed.

Lines 522-523: To achieve coastal disaster mitigation, accurate and efficient prediction (assessment) of storm surge is still an important task (see [8-13]).

Point 13: Line 507: you mention that you thoroughly reviewed the development of storm surge prediction. In fact, you mention some of the papers on storm surge prediction, but you did not critically review them. I also don’t think that this is useful in the framework of this paper but then you should not mention that you did a thorough review.

Response 13: Thanks for the suggestion. Indeed, we reviewed/summarized the development of storm surge prediction but did not make critical comments on those papers. Therefore, the key word “thoroughly” has been removed and the sentence has been rephrased.

Lines 525-527: The progress of storm surge modeling with different types of methodologies (i.e., empirical formulas, hydrodynamic models, and artificial intelligence approaches [2,15-16]) is reviewed and the remaining unresolved issues are addressed.

Point 14: Lines 557-564 is in my opinion not a very useful paragraph. I think it is too vague and fits more in a social science paper than in a hard core engineering paper. I would not include this.

Response 14: In the revised manuscript, the paragraph about future work (Lines 557-564) has been removed.

Point 15: On the other hand I think you did not mention one of the very big operational advantages of a good ANN model with a long-lead-time prediction. The model is very fast. You can easily make a dedicated model for many relevant locations along the Taiwanese coast. And on routine basis all these models can be run every hour to give every hour an updated surge prediction for the next 12 hours. Making the red curve of fig 10d can be done easily every hour. A hydrodynamic model can for the time being not do this on an hourly basis.

Response 15: The operational advantages of ANN models have been emphasized in the revised script.

Lines 109-111: Based upon fast computations, incorporation of ANNs into an operational storm surge forecasting system has also been reported [63].

Lines 516-519: A dedicated ANN model for different locations along Taiwan coastline can be easily built in a similar way. Based upon the operational advantages (i.e., fast computations) together with good capability for long-lead-time prediction, potentially, the newly-developed ANN model can be used in the near future for coastal disaster preparedness and early warning.

Lines 580-583: Success of the newly-developed ANN model with effective typhoon factors are proved/supported by satisfactory long-lead-time predictions. With great operational advantages (i.e., both of accuracy and efficiency for storm surge computation), potentially, the ANN model can be further utilized in coastal disaster management.
